# MAIT cells protect against pulmonary *Legionella longbeachae* infection

Huimeng Wang [1], Criselle D'Souza[1,2], Xin Yi Lim[1], Lyudmila Kostenko[1], Troi J. Pediongco[1], Sidonia B.G. Eckle [1], Bronwyn S. Meehan[1], Mai Shi[1], Nancy Wang[1], Shihan Li[1,3], Ligong Liu [4,5], Jeffrey Y.W. Mak [4,5], David P. Fairlie [4,5], Yoichiro Iwakura [6], Jennifer M. Gunnersen [7], Andrew W. Stent [8], Dale I. Godfrey [1,3], Jamie Rossjohn [9,10,11], Glen P. Westall[12], Lars Kjer-Nielsen[1], Richard A. Strugnell [1], James McCluskey [1], Alexandra J. Corbett [1], Timothy S.C. Hinks [1,13] & Zhenjun Chen[1]

Mucosal associated invariant T (MAIT) cells recognise conserved microbial metabolites from riboflavin synthesis. Striking evolutionary conservation and pulmonary abundance implicate them in antibacterial host defence, yet their functions in protection against clinically important pathogens are unknown. Here we show that mouse *Legionella longbeachae* infection induces MR1-dependent MAIT cell activation and rapid pulmonary accumulation of MAIT cells associated with immune protection detectable in immunocompetent host animals. MAIT cell protection is more evident in mice lacking $CD4^+$ cells, and adoptive transfer of MAIT cells rescues immunodeficient $Rag2^{-/-}\gamma C^{-/-}$ mice from lethal *Legionella* infection. Protection is dependent on MR1, IFN-$\gamma$ and GM-CSF, but not IL-17A, TNF or perforin, and enhanced protection is detected earlier after infection of mice antigen-primed to boost MAIT cell numbers before infection. Our findings define a function for MAIT cells in protection against a major human pathogen and indicate a potential role for vaccination to enhance MAIT cell immunity.

[1] Department of Microbiology and Immunology, Peter Doherty Institute for Infection and Immunity, University of Melbourne, Melbourne, VIC 3010, Australia. [2] Centre for Animal Biotechnology, Faculty of Veterinary and Agricultural Sciences, University of Melbourne, Melbourne, VIC 3010, Australia. [3] Australian Research Council Centre of Excellence in Advanced Molecular Imaging, University of Melbourne, Parkville, VIC 3010, Australia. [4] Division of Chemistry and Structural Biology, Institute for Molecular Bioscience, The University of Queensland, Brisbane, QLD 4072, Australia. [5] Australian Research Council Centre of Excellence in Advanced Molecular Imaging, The University of Queensland, Brisbane, QLD 4072, Australia. [6] Center for Animal Disease Models, Research Institute for Biomedical Sciences, Tokyo University of Science, Chiba-ken 278-8510, Japan. [7] Anatomy and Neuroscience Department, University of Melbourne, Melbourne, VIC 3000, Australia. [8] Faculty of Veterinary and Agricultural Sciences, University of Melbourne, Melbourne, VIC 3000, Australia. [9] Infection and Immunity Program and the Department of Biochemistry and Molecular Biology, Biomedicine Discovery Institute, Monash University, Clayton, VIC 3800, Australia. [10] Australian Research Council Centre of Excellence in Advanced Molecular Imaging, Monash University, Clayton, VIC 3800, Australia. [11] Institute of Infection and Immunity, Cardiff University, School of Medicine, Heath Park, CF14 4XN Wales, UK. [12] Allergy Immunology and Respiratory Medicine, Alfred Hospital, Melbourne, VIC 3004, Australia. [13] Respiratory Medicine Unit, Nuffield Department of Medicine Experimental Medicine and NIHR Oxford Biomedical Research Centre, University of Oxford, Oxfordshire, UK. These authors jointly supervised this work: Timothy S. C. Hinks, Zhenjun Chen. Correspondence and requests for materials should be addressed to J.M. (email: jamesm1@unimelb.edu.au)

Mucosal associated invariant T (MAIT) cells are innate-like lymphocytes with the potential to recognise a broad range of microbial pathogens. MAIT cells express a 'semi-invariant' αβ T-cell receptor (TCR) and recognise small molecule antigens presented by the major histocompatibility complex (MHC) class I-related molecule (MR1)[1,2]. These antigens comprise derivatives of the riboflavin biosynthetic pathway[3–5], which is conserved between a wide variety of bacteria, mycobacteria and yeasts[3,6], but is absent from mammals, and therefore provides an elegant mechanism to discriminate host and pathogen. Indeed, the enzymatic pathway required for riboflavin synthesis has been identified in all microbes shown to activate MAIT cells, and is absent in those that do not[3].

A striking feature of MAIT cell immunity is the high level of conservation of MR1 across 150 million years of mammalian evolution[7–9], implying a strong evolutionary pressure to maintain the MAIT cell compartment. Furthermore, MAIT cells have a strong pro-inflammatory phenotype[10] and are abundant in humans in blood and lung tissue[11], whilst in C57BL/6 mice they are found in greater abundance in the lungs than any other organ[12]. Together, these features implicate MAIT cells in a critical role in respiratory host defence. However, very few pathogens have been demonstrated in vivo to cause activation and proliferation of MAIT cells[13,14]. In studies implicating a role for MAIT cells in protective immunity against pathogens, the definition of these cells was limited by the lack of MR1-Ag tetramers[14]. To date, no studies have clearly defined a functional role for MAIT cells in protection against a clinically important human pathogen.

Using a model of bacterial lung infection with the intracellular bacteria Salmonella enterica serovar Typhimurium we have previously shown that riboflavin gene-competent bacteria can cause rapid activation and proliferation of MAIT cells[13]. We therefore hypothesised that this response could also be elicited with an authentic human lung pathogen and would contribute to protection against disease.

Legionella spp. are facultative intracellular pathogens, Gram-negative, flagellated bacteria which, when inhaled, cause a spectrum of disease from self-limiting Pontiac fever to severe, necrotic pneumonia: Legionnaires' disease[15]. The incidence of Legionnaires' disease has nearly trebled since 2000, with >5000 cases per year in the United States, inflicting a 10% mortality despite best treatment[16]. In North America and Europe[16] the predominant pathogen is L. pneumophila whereas in Australasia and Thailand more than 50% of cases are caused by L. longbeachae[17].

Here we use MR1 tetramers loaded with the potent MAIT cell ligand 5-(2-oxopropylideneamino)-6-D-ribitylaminouracil (5-OP-RU)[18] to identify[4] and characterise MAIT cells in human in vitro and mouse in vivo models of lung infection with the two most clinically significant Legionella species: L. pneumophila and L. longbeachae. Our data show that MAIT cells contribute to protection against fatal infection with Legionella, by a mechanism that is dependent on MR1, interferon-γ (IFN-γ) and granulocyte macrophage-colony stimulating factor (GM-CSF). Protection is partial in immunocompetent hosts but becomes increasingly evident as other arms of immunity are disabled such as in CD4 T cell-deficient animals. Protection ultimately becomes 'all or nothing' in profoundly immunodeficient mice RAG2$^{-/-}$γC$^{-/-}$ mice. These studies dissect the mechanisms by which MAIT cells contribute to protection against an important human disease and a model intracellular pathogen.

## Results

### *Legionella* activate human MAIT cells in vitro via MR1. We[3,13]
have previously shown that MAIT cells are activated by microbial species that express the riboflavin biosynthetic pathway; a finding which has been confirmed by others[6]. We therefore investigated whether *Legionella* species—*L. pneumophila*[15,19] and *L. longbeachae*[17]—known to cause serious pulmonary infections in humans and to express the necessary *rib* enzymes[20], could activate human MAIT cells. First, bacterial lysates of *L. pneumophila* and *L. longbeachae* stimulated a reporter cell line expressing a MAIT TCR (Jurkat.MAIT-A-F7)[3] in the presence of an MR1-expressing lymphoid cell line (C1R.MR1) (Fig. 1a, for gating strategy see Supplementary Fig. 1). Jurkat.MAIT cell activation was dose dependent, and could be specifically blocked by anti-MR1 antibody[21]. Next, we used a well-characterised human monocytic cell line (THP1.MR1)[22] as an antigen-presenting cell co-cultured with human peripheral blood mononuclear cells (PBMCs). We observed activation of MAIT cells when co-cultured with THP1 cells infected for 27 h with live *L. longbeachae* but not the co-cultured non-MAIT cells (Fig. 1b, c, Supplementary Fig. 2). Intracellular infection of wild-type THP1 and THP1.MR1+ cell lines induced expression of tumor necrosis factor (TNF) by human MR1-5-OP-RU tetramer+ MAIT cells. Activation was related to the infective dose, and was specific to MAIT cells and not non-MAIT CD3+ T cells. Activation was MR1 dependent, as it did not occur in the presence of cells in which we had disrupted the MR1 gene using a CRISPR/Cas9 lentiviral system (THP1.MR1−). MAIT cells also expressed IFN-γ in the presence of MR1-overexpressing cells (THP1.MR1+), but expression was less pronounced than TNF.

To visualise MAIT cells within their structural context in situ, we infected healthy human lung tissue ex vivo with *L. longbeachae* and observed CD3+ TCRVα7.2+ MAIT cells within the lung parenchyma in the proximity of free and intracellular *Legionella* bacilli 24 h post infection using immunofluorescence microscopy (Fig. 1d).

These findings indicate that *Legionella* induces significant MAIT cell immune responses in vitro and that MAIT cells are present at the site of infection in the vicinity of immune cells containing intracellular *Legionella* bacilli, suggesting that MAIT cells are likely to play a role in protection against *Legionella* pneumonia.

### *Legionella* Ag activates mouse MAIT cells in vitro. Next we
examined murine MAIT cell activation in vitro by *Legionella* using Vα19i Tg mice which have very high MAIT cell frequencies. We used a bone marrow-derived macrophage cell line transduced with the mouse MR1 gene (denoted as iBMDM.MR1) as antigen-presenting cells. As expected, we observed upregulation of the activation marker CD69 on transgenic splenic MAIT cells upon adding *Legionella* lysate. This activation was antigen-dose-dependent and partially MR1-dependent, being partially blocked by anti-MR1 antibody 8F2 but not isotype control (Fig. 2a). Interestingly, whilst anti-MR1 antibody completely blocked MAIT cell activation by the potent synthetic ligand 5-OP-RU, blockade of activation by lysate was incomplete, suggesting the lysate may contain some non-specific activating elements (e.g., Toll-like receptor (TLR) agonists) yet to be identified.

To visualise murine MAIT cells within their physiological context in situ we infected wild-type C57BL/6 (MAIT sufficient) and MR1$^{-/-}$ (MAIT deficient) mice with *L. longbeachae* and observed MR1-5-OP-RU tetramer$^+$, TCRβ$^+$ MAIT cells within the lung parenchyma of C57BL/6 mice (Fig. 2b), constituting the first demonstration of immunofluorescent staining of MAIT cells with murine MR1 tetramers. Importantly similar images were not observed in negative controls using the same tissue stained with non-specific control MR1-6-FP tetramer (Supplementary Fig. 3,

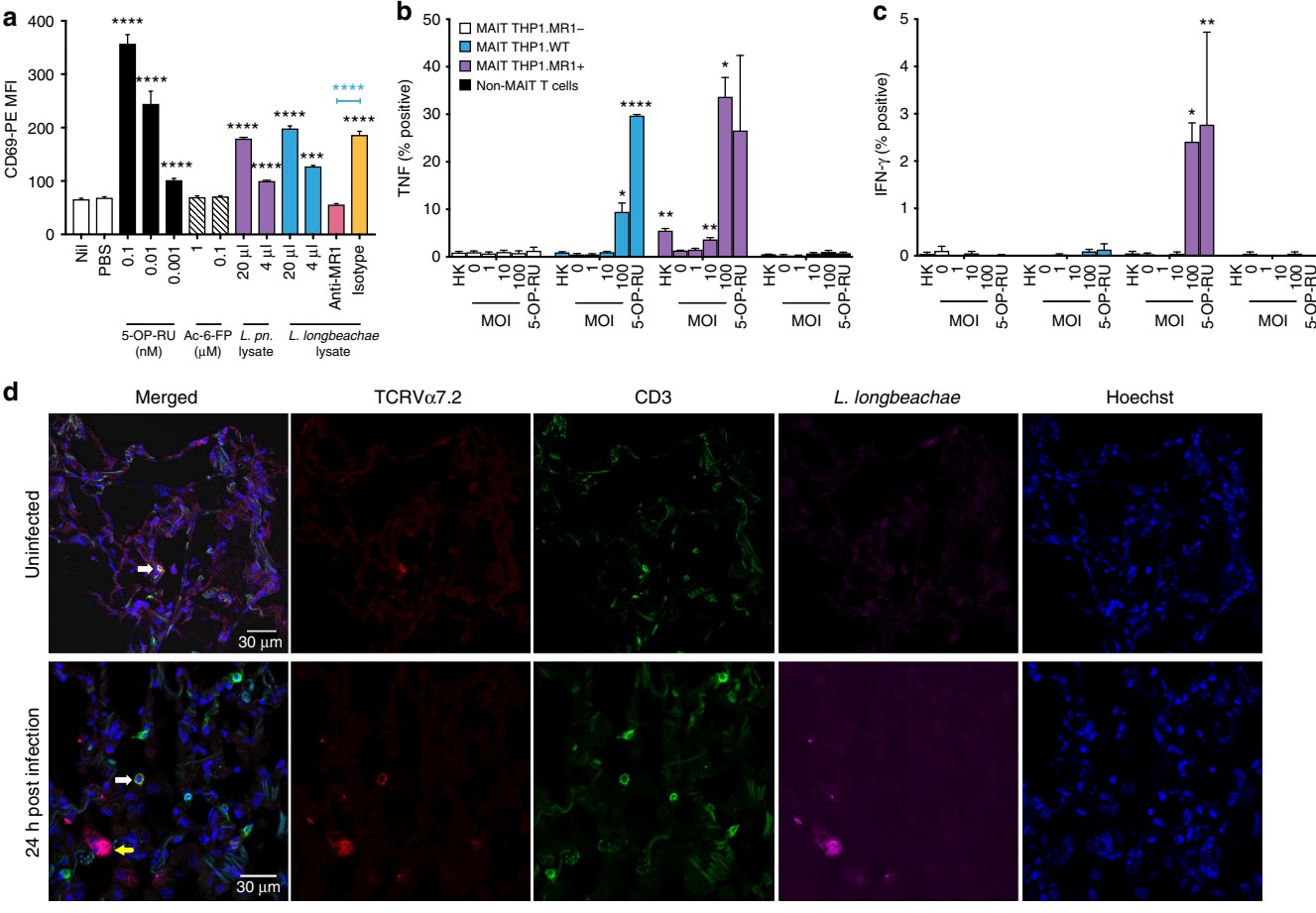

**Fig. 1** Human MAIT cells are activated by *Legionella* infection via MR1 in vitro. **a** Jurkat.MAIT and C1R.MR1 cells were co-incubated for 16 h with lysates of *L. pneumophila* (*L. pn.*) or *L. longbeachae* or 5-OP-RU, acetyl-6-formylpterin (Ac-6-FP) or PBS. Activation, detected by staining with anti-CD69, is enhanced by bacterial lysate or by the activating ligand 5-OP-RU, but not by acetyl-6-formylpterin. Activation was blocked by anti-MR1 antibody (26.5) but not by isotype control (W6/32) 2 h prior to co-incubation. Experiment performed in triplicate wells on two separate occasions with similar results. Data show mean fluorescence intensity, MFI (±SEM). Statistical tests: one-way ANOVA and post hoc Dunnett's comparing all columns with the first column (black). Unpaired *t*-test (blue), with ***$P < 0.001$; ****$P < 0.0001$. **b**, **c** THP1 cells (WT) or THP1 cells overexpressing MR1 (THP1.MR1+, purple) or deficient in expression of MR1 (THP1.MR1−, blue) were infected for 27 h with live or heat-killed (HK) *L. longbeachae* (MOI: 100) or 10 nM 5-OP-RU, then co-cultured for 16 h with sorted CD3+Vα7.2+CD161+ human peripheral blood MAIT cells, or MAIT-depleted conventional T cells. MR1-5-OP-RU-tetramer+ MAIT cell activation was measured by intracellular cytokine staining for **b** TNF or **c** IFN-γ. **b**, **c** Percentage cytokine-positive cells as mean (±SEM) data from three independent donors performed on two separate occasions are shown. Statistical tests: unpaired *t*-tests with Bonferroni corrections, each comparing against MOI 0 for the specific cell line. Statistics with *$P < 0.05$; **$P < 0.01$. **d** Immunofluorescence micrographs showing CD3+TCRVα7.2+ MAIT cell (white arrow) within healthy human lung tissue (top panel) and 24 h post infection (bottom panels) ex vivo with *L. longbeachae*. Yellow arrow: intracellular *L. longbechae* bacilli. Red, TCRVα7.2; green, CD3; magenta, polyclonal rabbit anti-*L. longbeachae*; blue, nuclei (Hoechst)

row 4) or infected MR1$^{−/−}$ mice (row 2). The low frequencies of MAIT cells in naive mice precluded similar co-stained cell images (row 1).

In light of the close parallels between in vitro human and murine MAIT cell studies, we undertook to investigate MAIT cell responses in mice in vivo.

**MAIT cells accumulate in lungs during *Legionella* infection.** Initially, we examined the impact of *Legionella* infection on MAIT cells (defined as CD45+TCRβ+ MR1-5-OP-RU tetramer-positive cells) in vivo in a murine model using intranasal (i.n.) infection with live *L. longbeachae* in immunocompetent mice. A small (0.4–2%) but distinct MAIT cell population can be seen in naive C57BL/6 mice (Fig. 3a, far left, for gating strategy see Supplementary Fig. 4). There was striking accumulation of pulmonary MAIT cells which comprised up to 30% of all pulmonary αβ-T cells after 7 days of infection (Fig. 3a–e). MAIT cell

accumulation was dependent on the initial infecting inoculum (Fig. 3a–c). The absolute number of MAIT cells increased up to 580-fold when infected with $10^5$ colony-forming units (CFU) ($P < 0.0001$, analysis of variance (one way-ANOVA) on log-transformed data with post hoc Dunnett's) *L. longbeachae*, compared with the much less dramatic increase in conventional αβ-T cells (maximum 9.4-fold, Dunnett's test $P < 0.0001$) (Fig. 3c). Accumulation occurred rapidly over 7 days post infection (DPI), with absolute numbers peaking around day 10 (Fig. 3d). Furthermore, despite a subsequent 20-fold contraction from peak frequencies ($P = 0.005$, Bonferroni-corrected *t*-test on log-transformed data), overall expansion of the MAIT cell population was long-lived, persisting for >280 DPI (Fig. 3d, e). Interestingly, although MAIT cells have been implicated in recruitment of non-MAIT T cells in a model of *Francisella tularensis* infection[14], we did not observe any significant difference in pulmonary recruitment of αβ-T cells in MR1$^{−/−}$ mice, which have an absolute deficiency of MAIT cells[12,13].

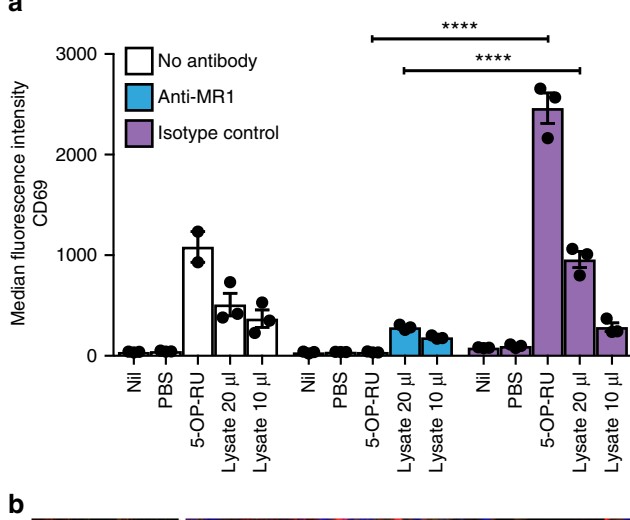

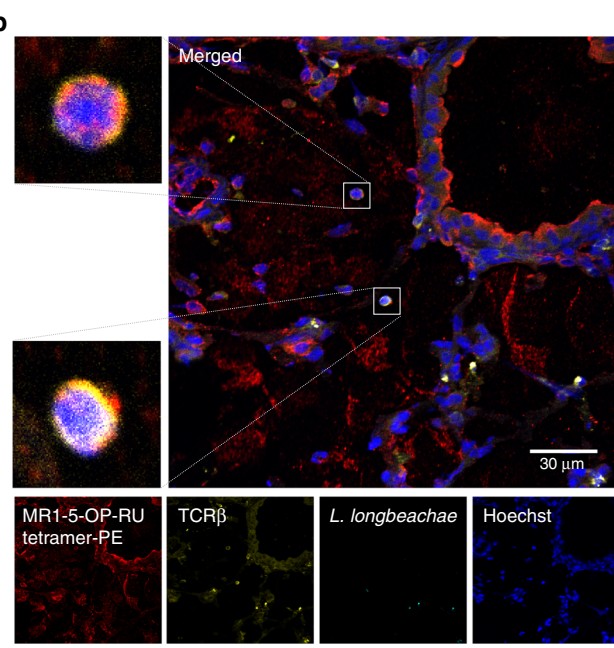

**Fig. 2** *Legionella* activates tetramer+ mouse MAIT cells in vitro via MR1. **a** Splenocytes ($2 \times 10^5$) from V$\alpha$19iC$\alpha^{-/-}$MR1+.Ly5.1 mice were prepared and co-cultured with iBMDM.MR1 cells ($10^5$) overnight with or without lysate (20 µl or 10 µl) of *L. longbeachae*. 5-OP-RU was used as a positive control at a final concentration of 1 nM. Anti-mouse MR1 monclonal antibody 8F2 or its IgG1 isotype control (in-house) were added at 20 µg ml$^{-1}$ to examine MR1-specific Ag presentation. CD69 upregulation (MFI) on MAIT cells was measured by flow cytometry for MAIT cell activation after staining together with other T-cell antibodies and MR1-5-OP-RU tetramers. Two way ANOVA with ****$P < 0.0001$. **b** Immunofluorescence micrographs of murine lungs showing TCR$\beta$+, MR1-5-OP-RU-tetramer+ MAIT cells (white boxes) in C57BL/6 mice infected intranasally with $2 \times 10^4$ CFU *L. longbeachae* 3 days prior. Images from uninfected or MR1$^{-/-}$ mice or stained with control tetramer MR1-6-FP are presented in Supplementary Fig. 3. Red, MR1-5-OP-RU tetramer; yellow, TCR$\beta$; cyan, *L. longbeachae*; blue, nuclei (Hoechst)

Like *L. longbeachae*, i.n. infection with $2 \times 10^7$ CFU *L. pneumophila* similarly induced a rapid expansion of MAIT cells (Supplementary Fig. 5), although more modest than *L. longbeachae*. As C57BL/6 mice are susceptible to *L. longbeachae*[17], *L. longbeachae* was selected as the most appropriate model for more detailed investigation.

Histology of lungs from mice infected with $2 \times 10^4$ CFU of *L. longbeachae* at 7 DPI demonstrated pronounced alveolar infiltration of neutrophils and macrophages, leukocytoclasia, aggregates of fibrin and accumulation of oedema fluid and epithelial shedding, consistent with the typical features of human *L. pneumophila* pneumonia[15] (Supplementary Fig. 6A, C). These inflammatory features were not observed in the uninfected mice (Supplementary Fig. 6B, D). Blinded analysis of these sections using a qualitative histological score at multiple time points post infection revealed inflammation peaked at day 7, but there were no gross histological differences in the severity of pneumonia between C57BL/6 and MR1$^{-/-}$ mice (Supplementary Fig. 6E). To determine the cellular localisation of *L. longbeachae* we measured bacterial burden in flow-sorted cells from collagenase-dispersed murine lungs 3 days post infection. Most viable bacilli localised within neutrophils, but evidence of infection of macrophages and dendritic cells was also observed (Supplementary Fig. 6F).

A number of clinical studies have reported decreases in peripheral blood MAIT cell frequencies associated with pulmonary infections[23–26], potentially attributable to recruitment from the blood. To address the question as to the source of MAIT cells at infection sites we used BrdU (5-bromo-2'-deoxyuridine) incorporation assay, which enables quantitation of DNA synthesis, reflecting cell proliferation. As expected, only a few BrdU+ MAIT cells (approximately 1%) and conventional T cells (about 1.3%) were enumerated in naive mice (Fig. 3f, far left, Fig. 3g) from both the lung and the draining mediastinal lymph nodes (dLNs). These low percentages represent the minimum background staining, and basal rates of self-renewal of pre-existing tissue resident memory cells[27]. As early as 3 DPI, an increased proportion of BrdU+ MAIT cells could be detected (lung, 4.8% and dLN, 2.4%) (Fig. 3f, g). On day 5 post infection, we detected a significant proportion of BrdU+ MAIT cells in both lung (26%) and dLN (48%) (Fig. 3f, g). Interestingly, the higher proportions of BrdU+ MAIT cells detected in the lungs compared with the dLNs at 3 DPI were reversed at 5 DPI (Fig. 3f, g). An equivalent change in these ratios was not observed among conventional T cells. This could indicate earlier activation and proliferation of MAIT cells locally in the lung tissue than in the dLN. Indeed, this is consistent with our previous work in which MAIT cell activation was detected as early as 2 h after antigen encounter, indicating lung MAIT cells could be activated without being primed in the draining LN[13]. Taken together, these data suggest that MAIT cells proliferate locally—both in the tissue and the dLN—upon infection. Furthermore, they suggest pulmonary MAIT cell proliferation commences earlier than in the dLN, consistent with the notion that conventional T-cell activation in the dLN requires additional time for the antigen-loaded dendritic cells to migrate to the dLN. The kinetics of conventional T cells were consistent with previous reports[28–31]. It is well recognised that naive T cells need 3–4 days before they are observed proliferating in the dLN and are observed a day later at the site of inflammation[28].

To explore MAIT cell function we investigated the dynamics of their cytokine profile throughout infection. During acute *L. longbeachae* infection, MAIT cells secreted interleukin (IL)-17A, IFN-$\gamma$ and GM-CSF (Fig. 4a, Supplementary Fig. 7A, B). The percentage of IL-17A-expressing MAIT cells was high and increased slightly throughout the course of infection (naive 22%, 7 DPI 27% and >100 DPI 30%; Fig. 4b, left panel), whilst IFN-$\gamma$ secretion was proportionally less, but nevertheless was significantly higher during the acute infection than in cells from uninfected mice or after resolution (ANOVA with post hoc Tukey's test $P = 0.03$ and 0.0004 respectively, Fig. 4b, middle panel). Conversely, the percentage of GM-CSF-expressing MAIT cells was lowest during acute infection and peaked after

disease resolution ($P = 0.0004$ acute vs resolution, $t$-test). Absolute numbers of IL-17A-expressing MAIT cells increased 200-fold from baseline in acute infection and remained 27-fold increased even after resolution of infection (Fig. 4c, left panel). The greatest differences observed were in IFN-γ-secreting MAIT cells, which increased 300-fold in acute infection, contracting during resolution to 12-fold above baseline (Fig. 4c, middle panel). Numbers of GM-CSF-secreting MAIT cells were increased 89- and 41-fold during acute infection and resolution

(Fig. 4c, right panel). No changes were observed with isotype (Supplementary Fig. 8A, B).

These cytokine expression profiles correlated closely with shifts in expression of nuclear transcription factors associated with T helper type 1 or 17 (Th1 or Th17) differentiation. In uninfected mice most (81 ± 4%, mean ± SD) MAIT cells expressed the orphan nuclear receptor, retinoic acid–related orphan receptor γ (RORγt) alone: a master regulator of Th17 cell differentiation (Fig. 4d, e, Supplementary Fig. 7C, D), but were negative for T-

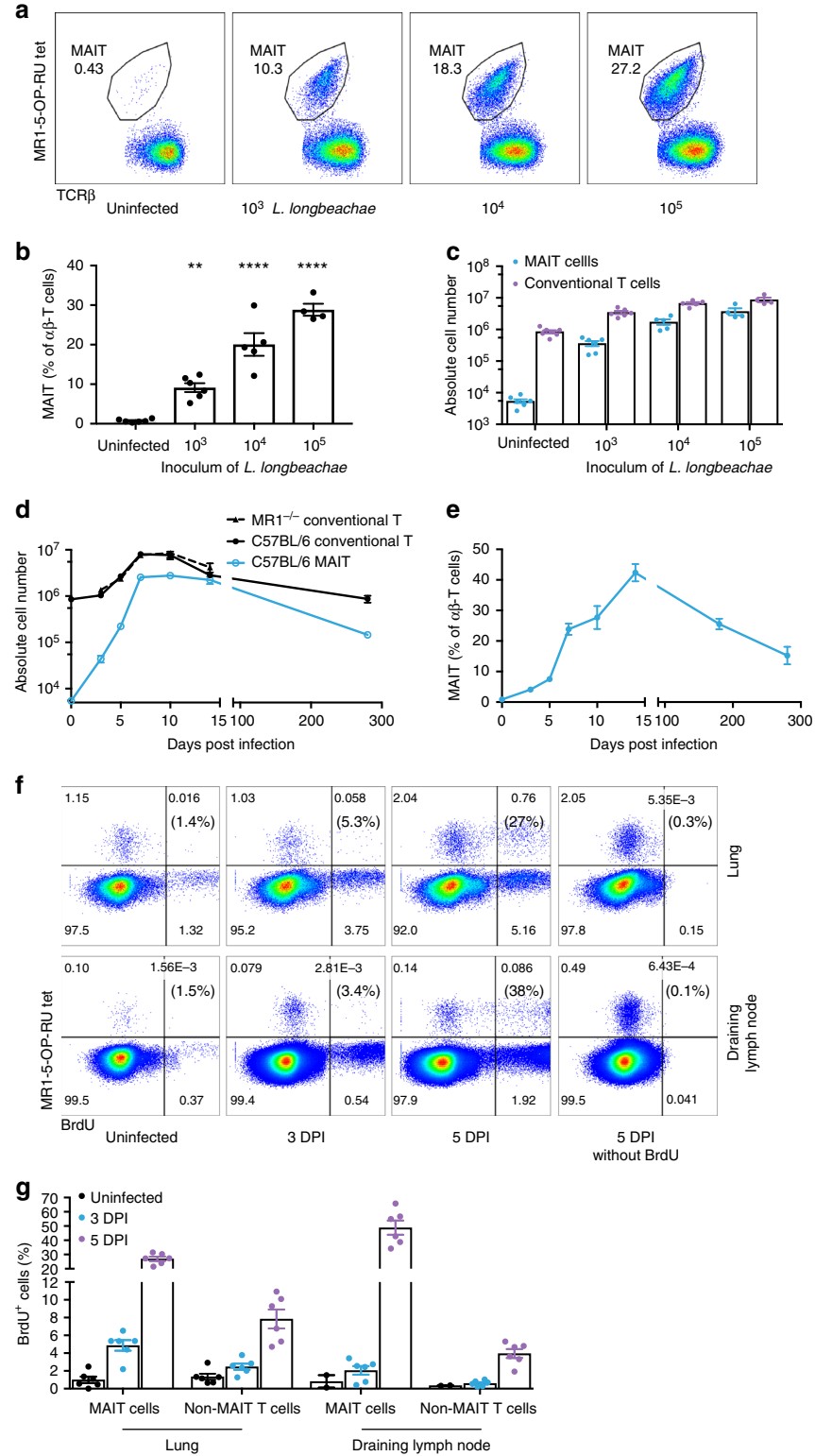

bet. Isotype controls are shown in Supplementary Fig. 8C. A minority (13 ± 4%) of cells expressed both RORγt and the Th1 regulator T-bet, and very few expressed T-bet alone. However, during acute infection and long-term post infection there was a marked shift in phenotype towards predominant co-expression of RORγt and T-bet in 64 ± 5% and 69 ± 3% of MAIT cells, respectively. MAIT cells expressing T-bet alone were only observed at significant frequencies (14 ± 3%) in acute infection. Thus, the consistent secretion of IL-17A in all stages of infection and the transient increase of IFN-γ secretion during acute infection reflect the changes in transcription factor profile we observed, suggesting the formation of an authentic memory pool of MAIT cells and pointing to a specific role for IFN-γ in the acute response to infection.

**MAIT cell protection against life-threatening *Legionella* infection is enhanced and accelerated by prior boosting.** To determine whether MAIT cells contribute to immune protection against *Legionella* we compared the bacterial burden in lungs of C57BL/6 and MR1$^{-/-}$ mice throughout infection. Bacterial load increased by 2.5 log over the initial inoculum, peaking at 3 DPI. In wild type versus MR1$^{-/-}$ on a C57BL/6 background we observed a significant difference in bacterial load, but not until days 10 and 14 post infection, immediately subsequent to the time of peak MAIT cell numbers at 7–9 DPI. Differences in bacterial load were of the order of one log in CFU, consistent with relatively impaired bacterial clearance in MAIT cell-deficient, MR1$^{-/-}$ mice (Fig. 5a, b).

In specific pathogen-free C57BL/6 mice, baseline frequencies of MAIT cells are very low[12,13], potentially due to a lack of natural exposure to diverse environmental pathogens. We have previously shown that MAIT cells can be expanded in vivo by intranasal exposure to the MAIT cell ligand 5-OP-RU with a TLR agonist such as the TLR9 agonist CpG or TLR2 agonist *S*-[2,3-bis (palmitoyloxy)propyl] cysteine (Pam2Cys) to furnish a MAIT cell costimulus[13]. To understand whether MAIT cell vaccination might impact on protection observed against *Legionella* infection of the lung, we used this approach to specifically expand pulmonary MAIT cells 1 month prior to *Legionella* infection, without affecting conventional T-cell frequencies (Fig. 5c, d). Prior exposure to 5-OP-RU and CpG enhanced MAIT cell numbers in the lungs and was associated with protection against infection as reflected in a reduction in bacterial load in C57BL/6 versus MR1$^{-/-}$ mice (Fig. 5e). This protective effect became apparent earlier than observed in wild-type C57BL/6 mice with reduced CFU seen on days 5 and 7 post infection and comparable on day 10 post infection (compare Fig. 5a, b). When a direct comparison was made between MR1$^{-/-}$ mice, C57BL/6 mice and C57BL/6 mice that had been boosted by 5-OP-RU and Pam2Lys, the bacterial burden was significantly lower on days 5, 7 and 10 post infection in wild-type mice that had received prior MAIT cell boosting (Fig. 5f). This demonstrates the potential to augment

MAIT cell-mediated protection by the prior administration of synthetic ligands as a 'vaccine'.

Studies of other intracellular pathogens have demonstrated high levels of functional redundancy in the ability of different lymphocyte subsets to control bacterial growth in vivo[32]. We hypothesised that by removing partially redundant effects of other lymphocyte subsets, the protective effects of MAIT cells would become more apparent. CD4+ T cell-derived IFN-γ has been shown to play an essential role in achieving bacterial clearance of *Salmonella* Typhimurium[32]. We therefore used GK1.5 transgenic mice, which express an anti-CD4 antibody and are CD4+ T-cell deficient[33], and compared these with GK1.5. MR1$^{-/-}$ mice which lack both CD4+ cells and MAIT cells. As expected, we observed a protective effect of MAIT cells through reduced bacterial burden apparent even earlier in the course of infection than with wild-type mice (statistically significant by 7 DPI) (Fig. 5g).

Collectively, these data demonstrate that MAIT cells contribute significantly to *Legionella* protection in the context of an intact immune system and that this protection is more rapid and of greater magnitude when mice are first vaccinated to expand MAIT cells before infectious challenge. Furthermore, the data from GK1.5Tg and GK1.5Tg.MR1$^{-/-}$ mice suggested to us that the role of MAIT cells could be revealed further if other immune subsets were removed.

**MAIT protection is more apparent in immune deficient mice.** To further unmask the full potential of MAIT cells in protection, we removed additional layers of immunity by studying the impact of adoptively transferred MAIT cells into profoundly immuno-deficient *Rag2*$^{-/-}$γC$^{-/-}$ mice in which Rag2 and the common γ chain are deleted, leading to absence of T, B and natural killer (NK) cells and several cytokine signaling pathways. We first expanded pulmonary MAIT cells by i.n. inoculation of donor mice with *S*. Typhimurium BRD509, as previously described[13]. Flow-sorted pulmonary MAIT cells from these mice were then adoptively transferred into recipient *Rag2*$^{-/-}$γC$^{-/-}$ mice. After transfer, administration of anti-CD4 and anti-CD8 monoclonal antibodies was used to further deplete any residual contaminating conventional T cells (Fig. 6a). After adoptive transfer, MAIT cells expanded spontaneously to generate a stable population by two weeks (Figs. 6b and 7a).

Although anti-CD4 and anti-CD8 mAbs were administered to deplete conventional CD4+ and CD8+ T cells, this exerted only a minor impact on eventual MAIT cell frequencies (Fig. 7b, c). Proportions of CD4+, CD8+ and CD4$^-$8$^-$ MAIT cells remained largely unchanged (Chen et al.[13] and Fig. 7c). Moreover, this temporary administration of anti-CD4 and anti-CD8 mAbs had no impact on weight loss or survival kinetics of *Legionella*-infected *Rag2*$^{-/-}$γC$^{-/-}$ mice (Fig. 7d, e). Direct comparison with conventional CD4+ or CD8+ T cells showed that MAIT cells could protect mice from lethal challenge similarly to CD8+

**Fig. 3** *L. longbeachae* induces long-lasting expansion of pulmonary MAIT cells. **a** Flow cytometry plots showing MAIT cell percentage among TCRβ+ lymphocytes in the lungs of C57BL/6 mice either uninfected or infected with 10$^3$, 10$^4$ or 10$^5$ CFU *L. longbeachae* for 7 days. Relative (**b**) and absolute (**c**) numbers of MAIT cells and conventional αβ T cells 7 days post infection. Kinetics of MAIT cells during 300 days after intranasal infection with 2 × 10$^4$ CFU *L. longbeachae*: absolute numbers (**d**), and relative percentages (**e**) of MR1-5-OP-RU tetramer+ MAIT cells or conventional αβ-T cells in C57BL/6 and MR1$^{-/-}$ mice. **f** Representative flow cytometry plots and **g** summarised data from BrdU incorporation assays to detect cellular proliferation in lung and draining lymph nodes of uninfected mice or at days 3 and 5 after intranasal infection with 10$^4$ CFU *L. longbeachae*. Percentages in brackets represent the proportion of MR1-5-OP-RU tetramer+ MAIT cells with BrdU incorporation. Experiments used 4–6 (**a–e**) or 3 (**f**, **g**) mice per group (mean ± SEM) and were performed twice with similar results. Mice were infected as described above and injected i.p. with 2 mg BrdU in 200 µl volume. After 2 h, mice were killed for T-cell preparation. Cells were stained with a cocktail of antibodies and 5-OP-RU tetramers first, then permeabilised and fixed for intranuclear DNA staining with anti-BrdU antibody. B6 C57BL/6 mice, BrdU bromodeoxyuridine, CFU colony-forming units, DPI days post infection. Statistical tests: ANOVA followed by Dunnett's test: **P < 0.01, ****P < 0.0001; C all comparisons with the respective uninfected (naive) groups P < 0.0001 for Dunnett's on log-transformed data. See also Supplementary figures 5, 6

conventional T cells, and slightly less than CD4+ conventional T cells in terms of both survival kinetics and bacterial load (Fig. 7f, g). We also assessed $Rag2^{-/-}$ mice which lack T and B cells but have an intact NK population. These mice recover well from pulmonary *Legionella* infection ($2 \times 10^4$ CFU) (Fig. 7h), indicating a role for NK cell immunity. Thus, MAIT cells function as part of a complex immune response, and $Rag2^{-/-}\gamma C^{-/-}$ mice constituted an optimal recipient strain to reveal MAIT cell functions.

Strikingly, the presence of adoptively transferred MAIT cells was sufficient to rescue completely $Rag2^{-/-}\gamma C^{-/-}$ mice from fatal infection with $10^3$ CFU *L. longbeachae* (Fig. 6c, $X_2$ $P < 0.0001$) in the absence of other components of adaptive immunity. Using a higher inoculum ($10^4$ CFU) we observed this protection was reduced by blockade with anti-MR1 mAb, which was associated with significantly reduced survival ($X_2$ $P = 0.005$) and with increased bacterial load amongst surviving mice (*t*-test

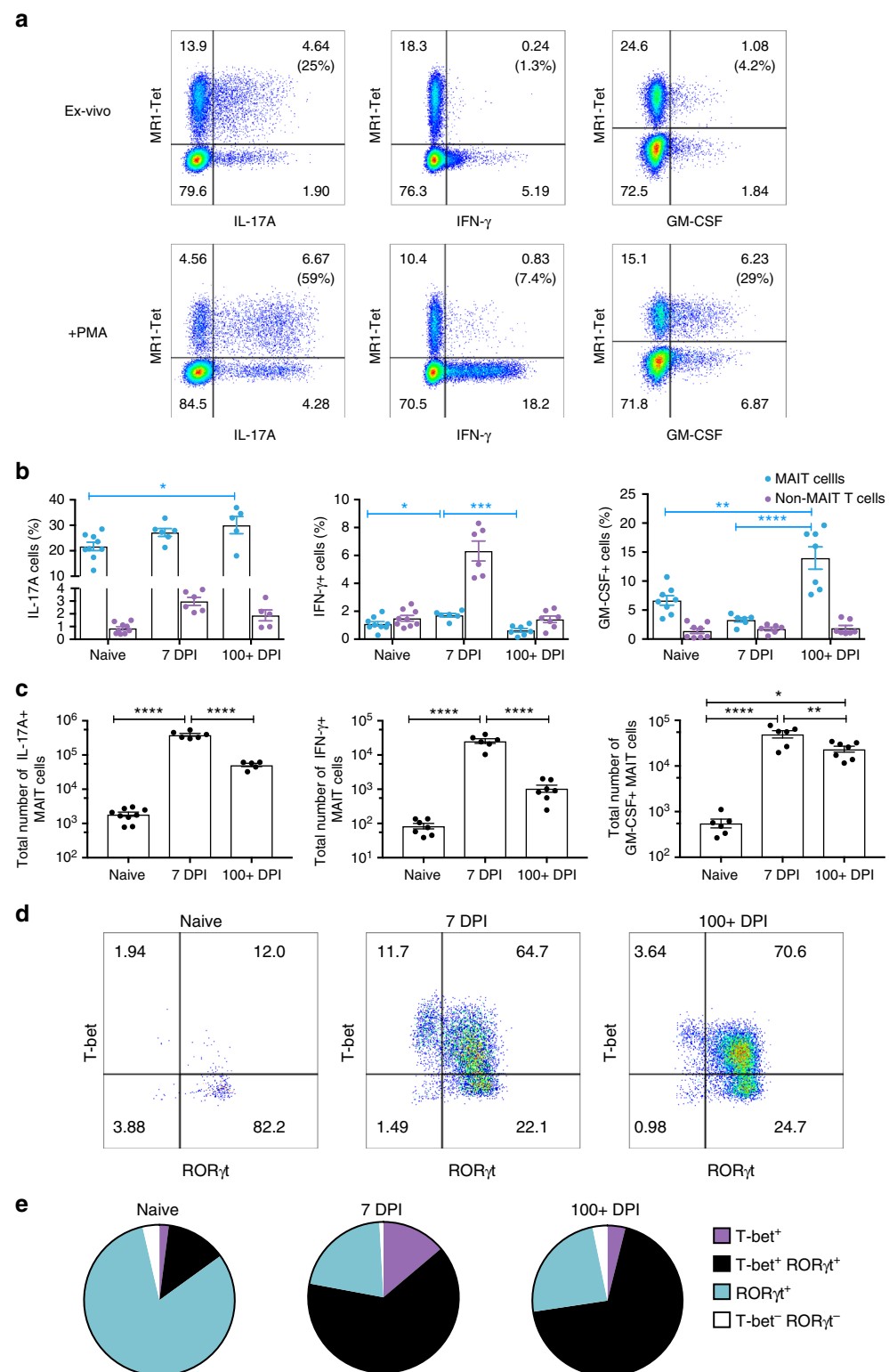

$P = 0.0004$), consistent with an MR1-dependent mechanism (Fig. 6d, e).

**MAIT cell protection is dependent on IFN-γ.** To determine the mechanism by which MAIT cells provide this protection we used adoptive transfer of MAIT cells from mice with deficiencies in cytotoxic capability or in pro-inflammatory cytokines. The protective effect of MAIT cells on both survival of $Rag2^{-/-}\gamma C^{-/-}$ mice or on bacterial burden was not impaired in MAIT cells lacking the cytolytic proteins perforin or granzymes A and B, nor in MAIT cells unable to express IL-17A or TNF (Fig. 8a, b). We observed a small increase in bacterial burden when transferred MAIT cells were deficient in GM-CSF (0.49 log-fold difference in CFU, $P = 0.026$), but this was not associated with significant differences in survival. By contrast, protection was critically dependent on MAIT cell-derived IFN-γ: when donor MAIT cells were deficient in IFN-γ, survival was decreased ($P < 0.0001$) and there was a 2.8 log-fold increase in bacterial burden (t-test $P < 0.001$) amongst the small proportion (3/21) of animals which survived till 23 DPI.In separate similar experiments, all mice transferred with IFN-γ deficient MAIT cells, succumbed within 35 days post *Legionella* infection.

Consistent with this protection, transferred MAIT cells expressed relevant pro-inflammatory cytokines (IL-17A, IFN-γ, GM-CSF) during *L. longbeachae* challenge. Adoptively transferred MAIT cells maintained an effector-memory cytokine profile in uninfected mice, similar to their corresponding 100 DPI counterparts, with the exception of GM-CSF (compare Fig. 4b with Fig. 8d). Most strikingly, MAIT cells increased IFN-γ expression significantly by >5-fold from 1.7% in C57BL/6 (Fig. 4a, b) to 9.8% (Fig. 8c, d), and GM-CSF expression increased 27-fold (3.3% to 9.1%), whilst IL-17A remained largely unchanged (27% to 30%).

The use of adoptive transfer of in vivo expanded MAIT cells provides compelling evidence that MAIT cells can confer protection against important human pathogens and demonstrates this protection depends upon their capacity to produce IFN-γ and to a lesser extent GM-CSF.

## Discussion

Our findings show that MAIT cells are activated and proliferate locally in response to *Legionella* infection, leading to enhanced immune protection in vivo that is dependent on IFN-γ and GM-CSF. This protection is evident earlier and of greater magnitude if mice are first vaccinated to expand and prime MAIT cells which are otherwise present in small numbers in normal mice. Protection by MAIT cells is characterised by more rapid reduction in bacterial loads and is MR1 dependent, suggesting mediation via antigen-specific activation. Remarkably, MAIT cell protection against *Legionella* was non-redundant and even evident in immune competent mice. The protective effect of MAIT cell immunity became more evident as layers of immunity were removed in host mice, firstly in GK1.5 mice lacking only CD4+ T cells and then in more

profoundly immunodeficient $Rag2^{-/-}\gamma C^{-/-}$ mice, lacking conventional T cells, B cells and NK cells. This observation is important given that studies of primary immunodeficiencies[34] imply redundancy of different lymphocyte subsets is a typical feature of pathogen immunity especially for innate mechanisms such as NK cells and innate lymphoid cells. Indeed, our T-cell transfer data in Fig. 7f, g and data from $Rag2^{-/-}$ mice in Fig. 7h suggested clear protective roles for CD4+, CD8+ T cells and NK cells. In the absence of B, T and NK cells in $Rag2^{-/-}\gamma C^{-/-}$ mice, MAIT cells were absolutely critical for survival in *Legionella*-infected mice, revealing their important potential in compromised hosts. As this mechanism is dependent on MR1, which presents small molecules derived from riboflavin biosynthesis[3–5], this demonstrates in vivo the potential for control of *Legionella* by detection of riboflavin metabolites.

The BrdU incorporation data demonstrated MAIT cell accumulation at the site of infection is largely due to local activation and proliferation. In several human studies[23,24,26], a decrease in MAIT cell numbers has been observed in the blood of patients with infections, including tuberculosis[25], and it has been hypothesised that MAIT cells traffic from the blood to the infected sites. We did not observe any fall in peripheral blood frequencies in our previous studies with *Salmonella* infection or vaccination[13,35]. It is still possible that in some infections inflammation-driven non-specific activation of MAIT cells[24,26,36] could lead to MAIT cell exhaustion and premature apoptosis. However, this speculation remains to be tested experimentally.

These observations suggest how the contribution of MAIT cells to immune protection may be critical to survival in clinical, naturally occurring severe infection. In essence, MAIT cells may be the difference between life and death in knife-edge infections where host immunity is partially compromised by comorbidities or predisposing factors, or where patients are exposed to large bacterial doses.

Although MAIT cells have both cytotoxic activity[37] and the ability to rapidly produce pro-inflammatory cytokines including IL-17A, TNF and IFN-γ[10], the protective effect of MAIT cells against *Legionella* infection was not dependent on TNF or IL-17A, but instead relied upon the capacity of MAIT cells to secrete IFN-γ and GM-CSF. This is consistent with a study of *Francisella* infection[38] where GM-CSF reduces bacterial burden late in the course of infection, although this did not translate into a significant survival difference. Nor did the protective effect depend on the key cytotoxic effector molecules: perforin and granzymes A/B. This is in contrast to work suggesting MAIT cell cytotoxicity is important for control of *Shigella*-infected HeLa cell lines in vitro[37].

As *Legionella* infects inflammatory cells, particularly neutrophils and macrophages, rather than epithelia, the essential immune function required of MAIT cells in our system is likely the IFN-γ-stimulated enhancement of bactericidal activity within these cells in which phagosome function has been subverted. Our findings of IFN-γ production upon MAIT cell activation are consistent with other reports[6,10,14,37], and accord with reports of

**Fig. 4** MAIT cell cytokine and transcription factor profiles during *Legionella* infection. **a** Representative flow cytometry plots showing intracellular staining for IL-17A, IFN-γ and GM-CSF by pulmonary TCRβ lymphocytes (non-MAIT conventional and MAIT cells) after 4 h of culture (direct ex vivo, top panel or with PMA/ionomycin stimulation, second panel) with brefeldin A. Lymphocytes were harvested from lungs of C57BL/6 mice infected with $2 \times 10^4$ CFU *L. longbeachae* for 7 days. Percentages in brackets represent the proportion of MR1-tetramer-positive MAIT cells expressing cytokine. **b** Percentages and **c** absolute numbers of pulmonary MAIT cells producing IL-17A, IFN-γ or GM-CSF by intracellular staining, directly ex vivo from C57BL/6 mice infected for 0, 7 or >100 days with $2 \times 10^4$ CFU *L. longbeachae*. Data were pooled with 5–9 mice per group (mean ± SEM) from 3 independent experiments. MAIT frequencies are compared using one-way ANOVA with post hoc Tukey's. *$P < 0.05$, **$P < 0.01$, ***$P < 0.001$, ****$P < 0.0001$. **d** Representative flow cytometry plots of MAIT cells expressing T-bet and RORγt. **e** Average proportion of T-bet+, RORγt+, double positive (DP) and double negative (DN) MAIT cells from uninfected or infected C57BL/6 mice at indicated date. Mean values of 5–9 mice in each group from 3 independent experiments. See also Supplementary figure 7C, 7D for individual data points. Mice were either naive or infected similarly as in **b**

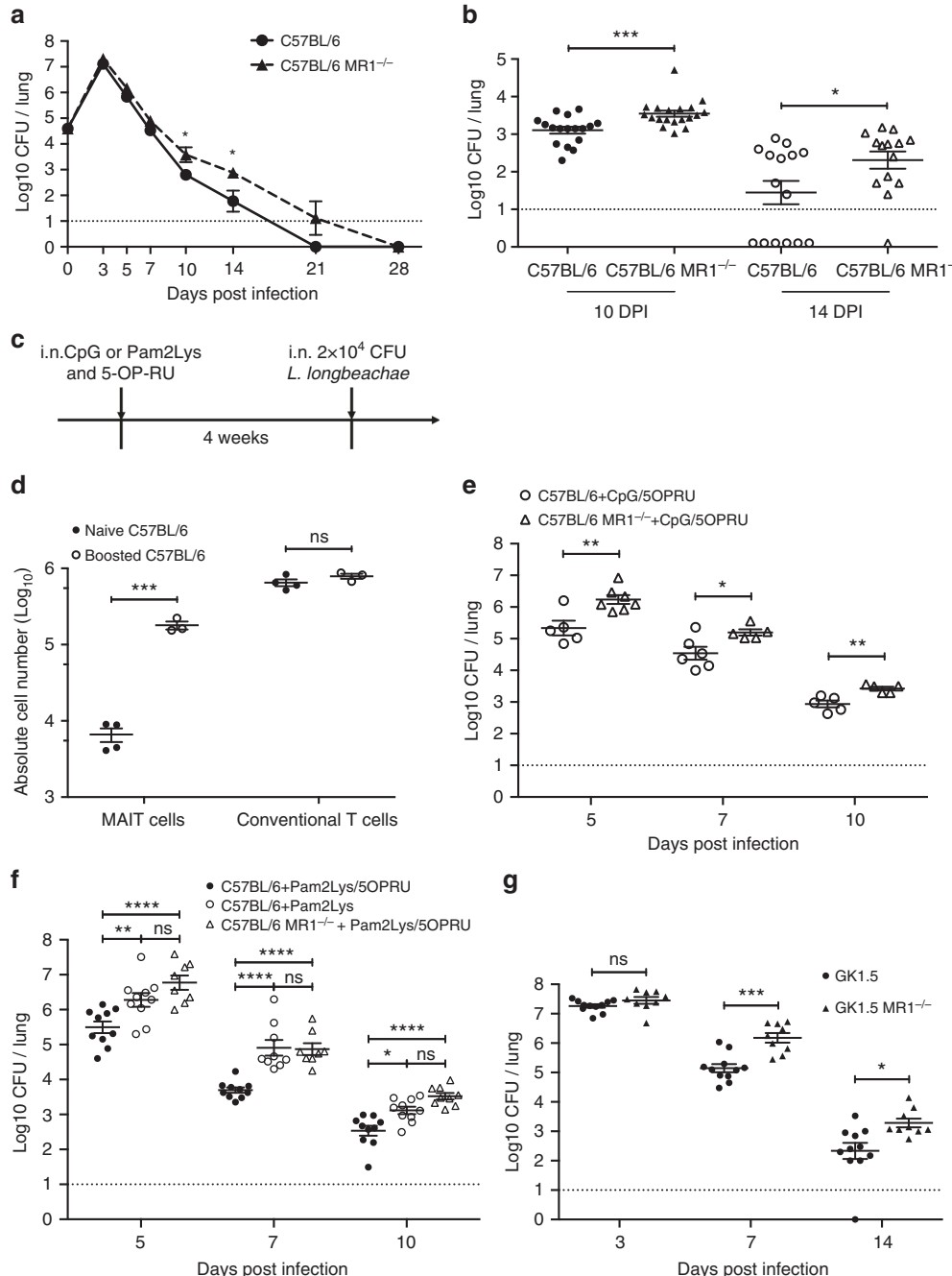

**Fig. 5** MAIT cell protection in *Legionella* infection accelerated by MAIT cell expansion. **a**, **b** Bacterial load (CFU) in lungs of C57BL/6 or MR1$^{-/-}$ mice following intranasal infection with $2 \times 10^4$ CFU *L. longbeachae*. Dashed line represents limit of detection. **a** Bacterial load over the time-course of infection. **b** Bacterial load at days 10 and 14 post-infection, from three independent experiments. Pooled data were compared using a Mann-Whitney test. **c** Schematic for **d**, **e**: C57BL/6 or MR1$^{-/-}$ mice were treated with 20 μg CpG (**d**, **e**) or 20 nmol Pam2Lys (**f**) in combination with 76 pmol 5-OP-RU (in 50 μl) intranasally 1 month before $2 \times 10^4$ CFU *L. longbeachae* inoculation. **d** Absolute numbers of MAIT cells and conventional T cells from lungs of naive or ligand-boosted C57BL/6 mice 30 days after administration of ligand. **e** Bacterial load comparison of vaccinated C57BL/6 vs MR1$^{-/-}$ mice. **f** Differences in bacterial load in lungs of C57BL/6 or ligand-boosted C57BL/6 or MR1$^{-/-}$ mice apparent at 5, 7 and 10 DPI. **g** Bacterial load in lungs of mice lacking CD4+ cells (GK1.5Tg) or CD4+ and MAIT cells (GK1.5Tg.MR1$^{-/-}$) 3, 7 and 14 days after infection with $2 \times 10^4$ CFU i.n. *L. longbeachae*. Pooled data (mean ± SEM) from two experiments with similar results using 5–6 mice per group, compared using two way-ANOVA tests on log-transformed data. *$P < 0.05$; **$P < 0.01$; ****$P < 0.0001$

a role for MAIT cell-derived IFN-γ in limiting growth of *Francisella tularensis* in bone marrow-derived macrophages in vitro[14]. IFN-γ has also been shown to enhance bactericidal activity of neutrophils via multiple mechanisms including enhancement of oxidative burst, nitric oxide production, antigen presentation, phagocytosis and upregulation of CD80/86 co-stimulation and T

cell-recruiting cytokines and chemokines[39]. Furthermore, T cell-derived IFN-γ has been shown to stimulate the bactericidal activity of monocyte-derived cells in murine *L. pneumophila* infection[40], and IFN-γ has been shown to be important for the control of *L. longbeachae*[41]. Moreover, given that IFN-γ is also critical for protection against mycobacterial disease including

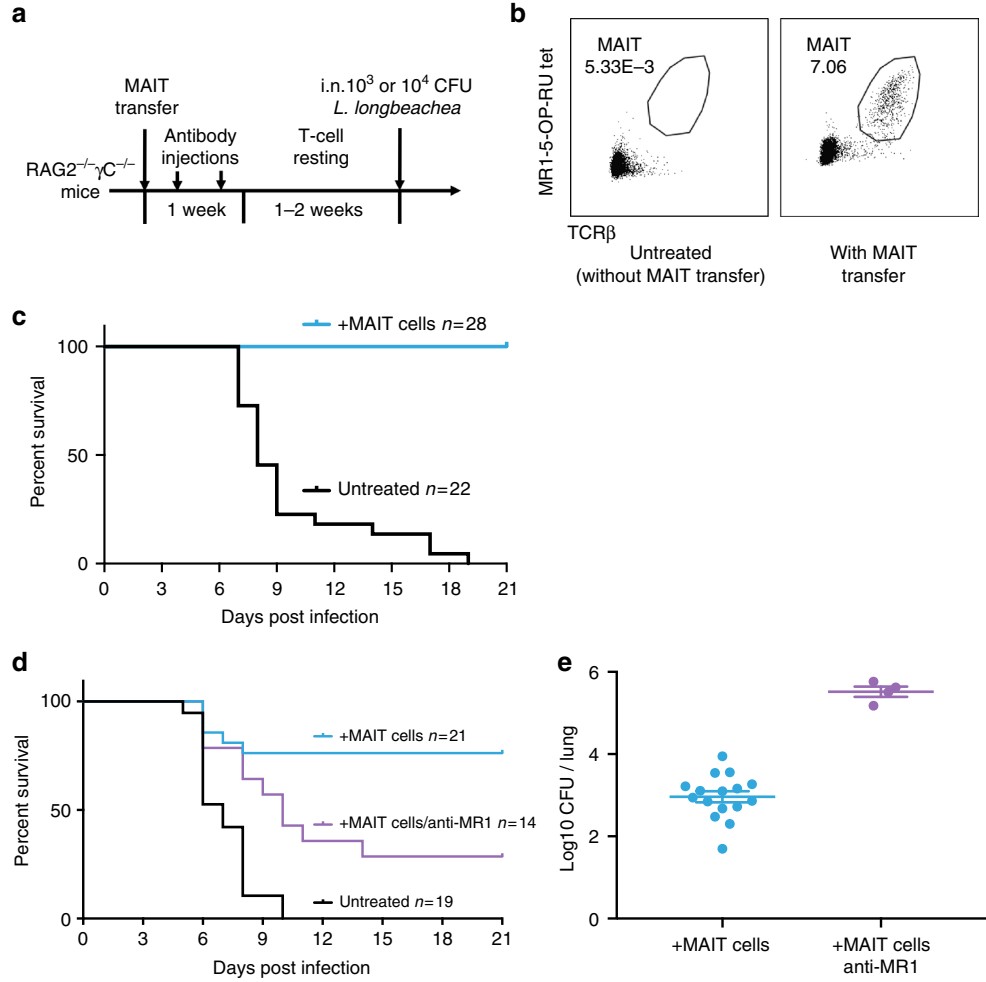

**Fig. 6** Adoptive MAIT cell transfer rescues $Rag2^{-/-}\gamma C^{-/-}$ mice from fatal *Legionella* infection. **a** Schematic of protocol: $10^5$ pulmonary MAIT cells from C57BL/6 mice previously infected with $10^6$ CFU *S.* Typhimurium BRD509 for 7 days to expand the MAIT cell population were sorted and transferred intravenously into $Rag2^{-/-}\gamma C^{-/-}$ mice, followed by intraperitoneal anti-CD4 and anti-CD8 antibody injection (0.1 mg each) twice within 1 week to deplete contaminating conventional T cells. After 2 weeks, mice were infected with $10^3$ (**c**) or $10^4$CFU (**d**) i.n. of *L. longbeachae*. **b** Representative plots showing live (7AAD−) hematopoietic (CD45.2+) cells with percentages of MAIT cells in the lungs of $Rag2^{-/-}\gamma C^{-/-}$ mice which were untreated or were recipients of adoptively transferred MAIT cells. **c** Survival of *Legionella*-infected untreated or MAIT cell-recipient $Rag2^{-/-}\gamma C^{-/-}$ mice after $10^3$ CFU i.n. *L. longbeachae* infection. **d** Survival of *Legionella*-infected untreated or MAIT cell-recipient $Rag2^{-/-}\gamma C^{-/-}$ mice after $10^4$ CFU i.n. *L. longbeachae*, with or without anti-MR1 blockade. One group received 0.25 mg anti-MR1 monoclonal antibody alternate days after infection. **e** Pulmonary bacterial load in surviving $Rag2^{-/-}\gamma C^{-/-}$ mice in **d** 23 DPI. Pooled data (mean ± SEM) from two independent experiments, each with 7–12 mice per group. See also Fig. 7

*Mycobacterium tuberculosis (M.tb)*, it is likely that this early production of MAIT cell-derived IFN-γ may be an important and non-redundant component of protection against mycobacteria. Indeed, in vitro MAIT cell-derived IFN-γ inhibits growth of *Bacillus Calmette-Guerin* (BCG) in macrophages[42], and the MR1–MAIT cell axis has been linked to susceptibility to BCG in mice[42] and to *M.tb* in humans[43] and mice[44].

A striking feature of MAIT cell biology is the very low frequencies of MAIT cells we observe in blood or lungs in naive mice[12,13], in contrast to the marked and long-lived expansion induced by a single infection in our model. Although antigen-naive MAIT cells have some intrinsic effector capacity[45], the delay between initial microbial exposure and peak MAIT cell frequency may be critical in providing a window of opportunity for a pathogen to exploit[14]. This notion is consistent with our observation that MAIT cell protection can be accelerated by prior expansion of the pulmonary MAIT cell population using intra-nasal synthetic 5-OP-RU and an appropriate TLR agonist. Notably, MAIT cell frequencies are low in early childhood[46], suggesting the potential to enhance the immunogenicity of

vaccines given in early life by incorporating such MAIT cell ligands in combination with TLR stimulation, which might for instance promote the recruitment of inflammatory monocyte differentiation via MAIT cell-derived GM-CSF[38].

In contrast, however, to a previous report that the protective effect of MAIT cells is mediated by enhanced recruitment of inflammatory monocytes to recruit more CD4+ T cells to the site of infection[38], in the GK1.5Tg vs GK1.5Tg.MR1$^{-/-}$ setting, in which there are no CD4+ T cells present, we nonetheless observed differences in bacterial burden in suggesting that MAIT cells can also contribute directly to immune protection.

MAIT cells show remarkable evolutionary conservation in several aspects: their conserved semi-invariant T-cell receptor, their shared antigen specificity across pathogen species and their restriction by MR1 molecules which are highly conserved across mammals. Our findings with MAIT cell priming demonstrated that previous immunisation not only generated a long-lasting expansion in MAIT cell frequencies (Fig. 3e), but also modified their cytokine profile (Fig. 4), affecting the quality of their effector functions. These effects together likely contributed to the

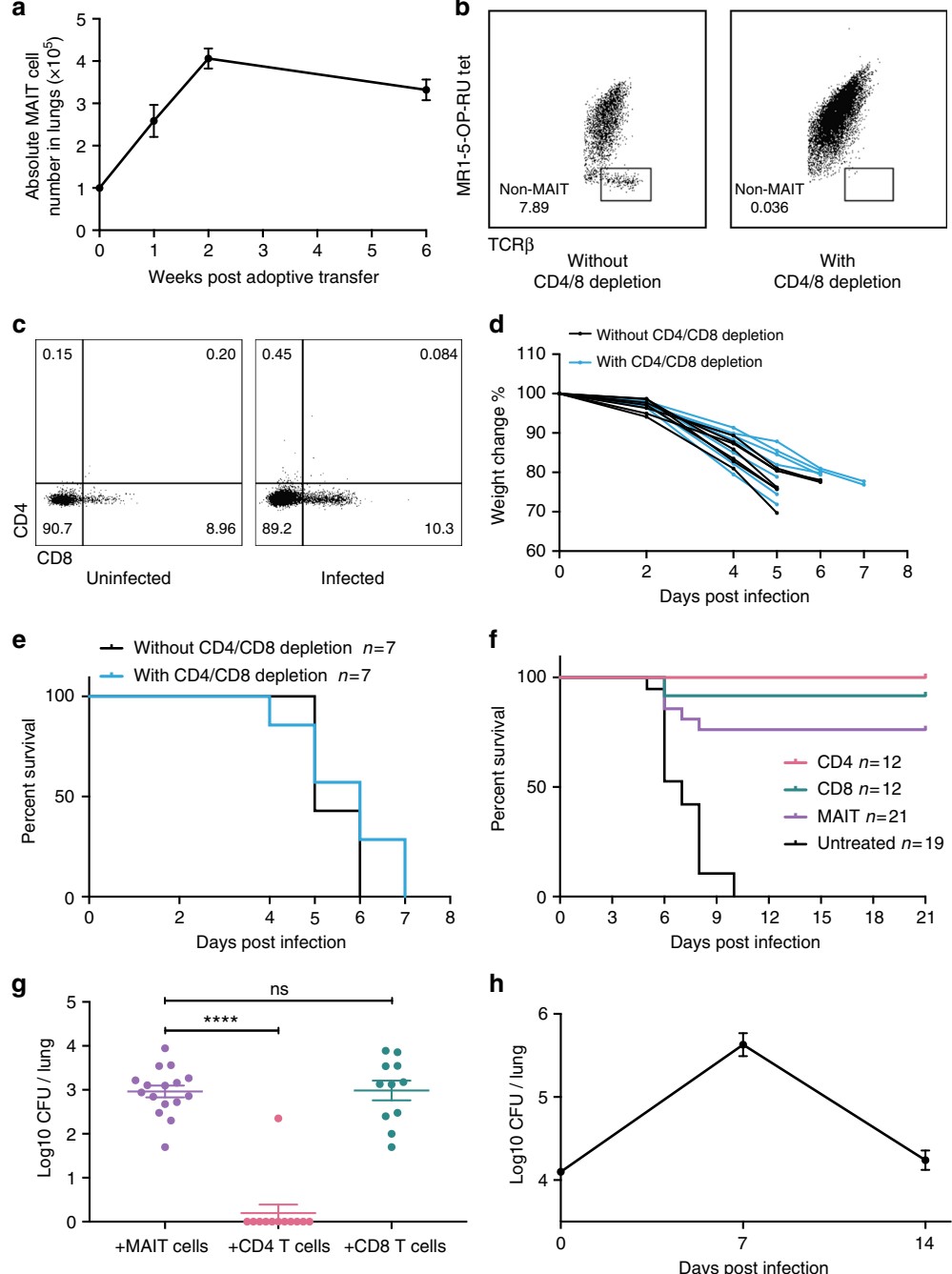

**Fig. 7** MAIT cell population expands after adoptive transfer. **a** Numbers of pulmonary MAIT cells increase in the lungs of recipient $Rag2^{-/-}\gamma C^{-/-}$ mice after i.v. adoptive transfer from mice infected with *S.* Typhimurium BRD509. **b** Residual contaminating conventional CD4+ or CD8+ T cells (<1% immediately post sort) expanded to 8–10% at 2 weeks post transfer (left) or were depleted completely (right) in mice treated twice in the week post transfer with 0.1 mg each of anti-CD4 and anti-CD8 antibody i.p. **c** MAIT cell composition transferred to $Rag2^{-/-}\gamma C^{-/-}$ mice before and after infection with *L. longbeachae*. Profiles are similar to C57BL/6 mice, implying minimal impact of antibody depletion. **d** Weight change and **e** survival curves of $Rag2^{-/-}\gamma C^{-/-}$ mice with (blue) or without (black) anti-CD4 and anti-CD8 treatment, challenged with $10^4$ CFU *L. longbeachae* implying minimal impact of anti-CD4 and anti-CD8 antibodies on innate immunity in $Rag2^{-/-}\gamma C^{-/-}$ mice. **f** Direct comparison of protective potency of MAIT and conventional CD4+ or CD8+ T cells. Mice received $10^5$ transferred MAIT, CD4+ and CD8+ cells and were treated respectively with anti-CD4 and anti-CD8, anti-CD8 only and anti-CD4 only and then challenged with $10^4$ CFU *L. longbeachae*. **g** At 21 DPI, surviving mice were examined for bacterial load; $n = 12$–19 mice pooled from 2 independent experiments. **h** $Rag2^{-/-}$ mice infected with $2 \times 10^4$ CFU *L. longbeachae* were capable of clearing *Legionella*; $n = 5$ mice/time point. The experiment was performed twice with similar results

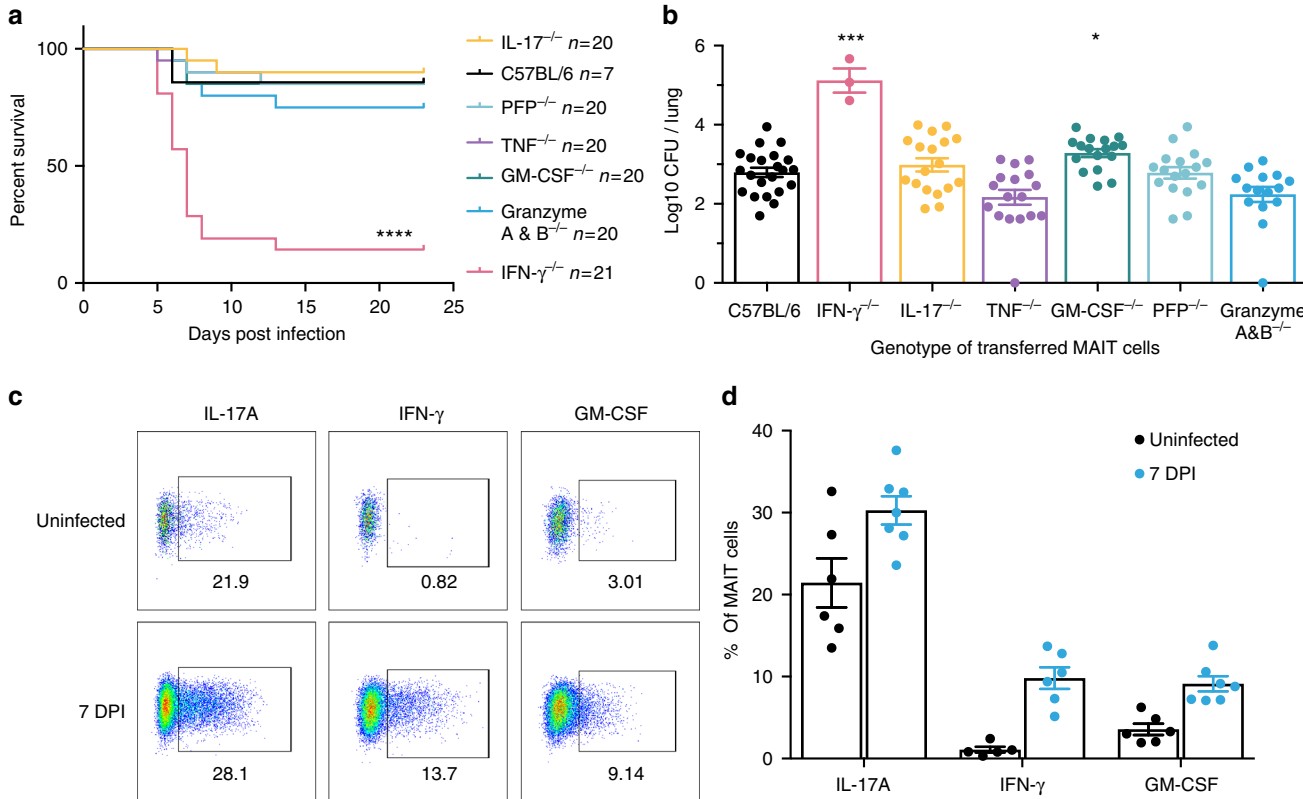

**Fig. 8** Protection of $Rag2^{-/-}\gamma C^{-/-}$ mice from *Legionella* is dependent on IFN-γ. **a** Survival of $Rag2^{-/-}\gamma C^{-/-}$ mice adoptively transferred with pulmonary MAIT cells generated from different mouse strains, after $10^4$CFU i.n. *L. longbeachae*. ****Gehan–Breslow-Wilcoxon $P < 0.0001$. **b** Pulmonary bacterial load in surviving $Rag2^{-/-}\gamma C^{-/-}$ mice in **a** 23 DPI. Pooled data (mean ± SEM) from two replicates with similar results, each with 7–13 mice per group, except the $Ifng^{-/-}$ MAIT transferred group, in which only 3 mice survived to 23 DPI. **c** Representative flow cytometry plots and **d** percentages of cytokine expressing transferred MAIT cells by intracellular staining (direct ex vivo) for IL-17A, IFN-γ and GM-CSF from uninfected (top panel) or infected (7 DPI, bottom panel) mice after 4 h of culture with brefeldin A. Percentages in brackets in **c** represent the proportion of MR1-tetramer-positive MAIT cells expressing cytokine. Data were pooled with 5–7 mice per group (mean ± SEM) from two independent experiments. Mean (**b**) is shown. PFP perforin. Bonferroni-corrected *t*-tests *$P < 0.05$; ***$P < 0.001$

enhanced protection observed (Fig. 5e, f). We believe this work is the first to demonstrate that MAIT cells form a memory pool. As it remains debatable whether invariant NKT cells are capable of forming a stable memory population[47], it therefore enriches our understanding of innate-like T cells, and provides a basis from which to explore therapeutic approaches, including vaccination. A protective effect of expanded MAIT cells could contribute to heterologous protection afforded by neonatal BCG vaccination against other, unrelated classes of pathogens[48] and could have implications for immunocompromised populations and in managing infections with multiple drug-resistant bacteria. Vaccination with MAIT cell ligands might help resolve chronic infections where MAIT cell frequencies may be reduced due to other therapies[11], comorbidities or activation-induced cell death[49].

Our findings define a significant role for MAIT cells in pulmonary host defence against a major human pathogen. We reveal layers of immunological redundancy likely to mask the contribution of MAIT cells in many situations of infectious challenge, but suggest a critical role for MAIT cells becomes apparent in a crisis situation—as reflected here in a high-infecting inoculum, or in the face of compromised specific immunity—in which the gulf between survival and death is finely balanced. Moreover, we demonstrate the mechanism of this MAIT cell protection is IFN-γ dependent and enhanced by GM-CSF. Due to the pleiotropic roles of IFN-γ and the conservation of the riboflavin pathway across many species, this mechanism is likely broadly effective

against other major human intracellular pathogens, and may prove as relevant to the later stages of infection as to the initial, acute phase characterised by innate immune responses countering rapid pathogen replication. We have shown this immunity can be augmented by exposure to MR1 ligands, suggesting this mechanism might have potential for preventive or therapeutic benefit.

## Methods

**In vitro activation assays.** Jurkat cells expressing a MAIT TCR comprising the TRAV1-2-TRAJ33 α-chain and TRBV6-4 β-chain (Jurkat.MAIT)[3] were co-incubated at $10^5$ cells per well in 96-well U-bottom plates with an equal number of class I reduced (C1R) antigen-presenting cells (APCs) expressing MR1 (C1R.MR1)[3] for 16 h in RPMI-1640 media (Gibco) in supplement and 10% foetal calf serum (RF10) media at 37 °C, 5% $CO_2$. Cells were stimulated for 16 h with bacterial lysates prepared using repeated ultrasonication of bacteria in log-phase growth. Cells were stained with anti-CD3-APC and anti-CD69-PE Abs and 7-aminoactinomycin D (7AAD; 1:500) before flow cytometric analysis. Activation of Jurkat.MAIT cells was measured by an increase in surface CD69 expression.

For human MAIT cell assays, PBMCs were stained with anti-CD3-PECF594, CD161-PECy7, TCRVα7.2 (TRAV1-2)-APC. Vα7.2+ cells were enriched with anti-APC magnetic beads and the CD3+Vα7.2+CD161+ population was isolated by flow cytometry and cultured at $10^4$ cells/well for 16 or 36 h in penicillin-free media containing streptomycin and gentamicin with $5 \times 10^4$ THP1 cells which had been first infected for 3 h (CD69 assays) or 27 h (intracellular cytokine staining) with different multiplicities of infection (MOIs) of live *L. longbeachae* NSW150. Control wells contained 5-OP-RU 10 nM or *L. longbeachae* NSW150 at MOI 100 which had been heat killed for 10 min at 67 °C. CD69 upregulation was measured by surface staining for CD69-APCCy7 and 5-OP-RU loaded human MR1 tetramer-PE. For intracellular cytokine expression of cells, brefeldin A was added for the last 16 h, and cells were fixed, permeabilised (using BD Fixation/Permeabilization Kit

(BD, Franklin Lakes, NJ) and stained with MR1 tetramer, CD3-PE CF594, Zombie yellow, IFN-γ-FITC and TNF-Pacific blue.

For murine MAIT cell assays, splenocytes ($1–2 \times 10^5$) from Vα19iCα$^{-/-}$MR1$^+$. Ly5.1 mice were prepared and co-cultured with iBMDM.MR1 cells ($10^5$) overnight with or without lysate of *L. longbeachae*. iBMDM.MR1 cells were used as APCs. 5-OP-RU was used as a positive control at final concentration of 1 nM. Anti-mouse MR1 mAb 8F2.F9 or its IgG1 isotype control (in-house) were added to examine MR1-specific antigen presentation. CD69 and CD25 upregulation or the percentage of CD69/CD25-expressing cells were measured by flow cytometry for MAIT cell activation after staining together with other T-cell antibodies.

**Generation of cell lines**. THP1.MR1− cell lines were generated by targeted deletion of MR1 using *Lentiviral* CRISPRv2 which was a gift from Feng Zhang (Addgene plasmid # 52961)[50]. The plasmid was digested with *BsmB*1 (Fermentas), dephosphorylated and purified using gel electrophoresis and the Ultraclean DNA isolation kit (MO Bio Laboratories, Carlsbad CA). Short guide RNAs (ACCTCTCATCATTGTGTTAA) were ligated and the reaction product used to transform Stbl3 *Escherichia coli*. DNA was purified from transformed colonies (QIAprep spin miniprep) and used to transfect HEK293T (ATCC CRL-3216, ATCC, Manassas, VA) cells with LentiviralCRISPRv2 vector and packaging vectors. Supernatants were used to transduce parent THP1 cells (ATCC TIB-202, ATCC) and cells were selected using puromycin resistance and single-cell sorting using anti-MR1 antibody (8F2.F9) after upregulation of MR1 using acetyl-6-FP. MR1 knockouts were then verified using surface staining and western blotting.

C1R.hMR1, THP.hMR1+ and iBMDM.MR1 cell lines were created via retroviral transduction as described previously[51], using human or murine MR1 cloned in pMIG plasmids. BMDMs were commercially available (EMG014, Kerafast, Boston, MA). At the end of the transduction, MR1 expression of the corresponding cell lines was analysed by staining with anti-MR1 antibodies (26.5 for hMR1 and 8F2 for mMR1) by fluorescence-activated cell sorting (FACS). In these transduced cell lines, green fluorescent protein (GFP) expression levels correlated well with surface expression of MR1. Cells with high expression of both GFP and MR1 were enriched by FACS and subsequently cloned by single-cell sorting using flow cytometry. The function of tranduced MR1 were confirmed by in vitro cellular activation assays using Jurkat.MAIT as previously described[13].

**Immunofluorescence microscopy**. The 8 μm sections of cryopreserved, unfixed human and murine lung tissue were submerged into ice-cold acetone for 10 min, air dried and then re-hydrated in PBS for 10 min. Endogenous biotin block was performed using Biotin/Avidin blocking kit (Thermo Fisher, Waltham MA) according to the manufacturer's instructions. Serum-free protein block (DAKO, Carpinteria, CA) was applied for 15 min, followed by 30 min of blocking with 10% normal donkey serum. Sections were subsequently blocked with Murine MR1-6-FP tetramer (nil conjugate) for 1 h at room temperature. Murine MR1-5-OP-RU tetramer-PE (25 μg ml$^{-1}$ in 2% bovine serum albumin/PBS) was applied for 1 h at room temperature in the dark, sections washed with PBS, fixed with 1% paraformaldehyde for 10 min, washed again and stained with a cocktail containing polyclonal rabbit anti-*Legionella* antibody (gift from Dr Hayley Newton, Department of Microbiology and Immunology, University of Melbourne) and Goat anti-PE (KPL). After 1 h, sections were washed and stained with Donkey anti-Goat-Alexa Fluor 568 (Life Technologies), Donkey-anti-Rabbit-DyLight 680 (Life Technologies) and Alexa Fluor 647-conjugated Rat anti-mouse TCRβ (Biolegend). Nuclei were counterstained with Hoechst 33342 (Life Technologies).

Following rehydration, the unfixed cryopreserved 8 μm sections of human lung were blocked with serum-free block and stained with purified anti-Vα7.2 TCR mAb clone 3C10 (Biolegend), rat anti-human CD3 mAb (BioRad) and polyclonal rabbit anti-*Legionella* antibody. After 1 h at room temperature, sections were washed and stained with a cocktail of secondary antibodies containing goat anti-mouse Alexa Fluor 568, goat anti-rat Alexa Fluor 488 and goat anti-rabbit Alexa Fluor 647, all from Life Technologies. Nuclei were counterstained with Hoechst 33342 (Life Technologies).

Sections were mounted with Prolong Gold mounting medium (Life Technologies). Image acquisition was performed on Zeiss LSM 710 confocal microscope using Zen software (Zeiss, Oberkochen, Germany) The resultant images were further analysed using FIJI Image J software[52] (UW-Madison).

**Mouse models**. Mice were bred and housed in the Biological Research Facility of the Peter Doherty Institute (Melbourne, Victoria, Australia). MR1$^{-/-}$ and Vα19iCα$^{-/-}$MR1$^{+/+}$ (Ly5.1) mice were generated by breeding Vα19iCα$^{-/-}$MR1$^{-/-}$ mice[53] (from Susan Gilfillan, Washington University, St Louis School of Medicine, St Louis, MO) with C57BL/6, Ly5.1 and TCRα$^{-/-}$ mice separately and inter-crossing of F1 mice. The genotype was determined by tail DNA PCR at the MR1 locus or flow cytometry after staining with relevant antibodies as previously described[13]. Granzyme A$^{-/-}$, B$^{-/-}$ or double knockout (KO; A$^{-/-}$, B$^{-/-}$) and Perforin$^{-/-}$ mice were provided by Joe Trapani (Peter MacCallum Cancer Centre, Melbourne). GK1.5 mice were crossed onto the MR1$^{-/-}$ background to generate GK1.5.MR1$^{-/-}$ mice which lack CD4$^+$ cells and MAIT cells. Specific pathogen-free, co-housed male mice aged 6–12 weeks were used in experiments, after approval by the University of Melbourne Animal Ethics Committee (1513661).

**Intranasal infection**. The i.n. inoculation with *L. longbeachae* or antigens (76 pmol 5-OP-RU) and TLR agonist (either 20 μg CpG or 20 nmol Pam2Cys) in 50 μl per nare was performed on isofluorane-anaesthetised mice. For blocking experiments, mice were given 250 μg anti-MR1 (26.5 or 8F2.F9)[21,54] or isotype control mAbs in 200 μl PBS, once intraperitoneally (i.p.) 1 day prior to inoculation and three times (d1, d3, d5) post inoculation. Mice were killed by CO$_2$ asphyxiation, perfused through the heart with 10 ml cold RPMI and lungs were taken.

To prepare single-cell suspensions, lungs were finely chopped with a scalpel blade and treated with 3 mg ml$^{-1}$ collagenase III (Worthington, Lakewood, NJ), 5 μg ml$^{-1}$ DNAse and 2% foetal calf serum in RPMI for 90 min at 37 °C with gentle shaking. Cells were then filtered (70 μm) and washed with PBS/2% foetal calf serum. Red blood cells were lysed with hypotonic buffer TAC (Tris-based amino chloride) for 5 min at 37 °C. Approximately $1.5 \times 10^6$ cells were filtered (40 μm) and used for flow cytometric analysis.

**Bacterial counts in infected lungs**. Bacterial colonisation was determined by counting CFU obtained from plating homogenised lungs in duplicate from infected mice (×5 per group) on buffered charcoal yeast extract agar containing 30 μg ml$^{-1}$ streptomycin and colonies counted after 4 days at 37 °C under aerobic conditions.

**Adoptive transfer**. As MAIT cell frequencies are low in naive C57BL/6 mice, prior to adoptive transfer experiments, MAIT cell populations were expanded by i.n. infection with $10^6$ CFU S. Typhimurium BRD509 in 50 μl PBS for 7 days as previously described[13]. After 7 days, mice were killed, single-cell suspensions prepared and live CD3+CD45+MR1-5-OP-RU tetramer+ cells sorted using BD FACS Aria III. Next, $10^5$ MAIT cells were injected into the tail veins of recipient mice which then received 0.1 mg each of anti-CD4 (GK1.5) and anti-CD8 (53.762) intraperitoneally on days 2 and 5 or 6 to control residual conventional T cells. Mice were rested for 2 weeks post transfer to allow full expansion of the MAIT cell population prior to subsequent infectious challenge. Mice were weighed daily and assessed for visual signs of clinical disease, including inactivity, ruffled fur, laboured breathing and huddling behaviour. Animals that had lost ≥20% of their original body weight and/or displayed evidence of pneumonia were killed.

**Statistical analysis**. Statistical tests were performed using the Prism GraphPad software (version 7.0 La Jolla, CA). Comparisons between groups were performed using Student's t-tests, ANOVA tests or Mann–Whitney tests as appropriate unless otherwise stated. Survival curves were compared using the Gehan–Breslow–Wilcoxon method for multiple groups.

**Reagents**. Human PBMCs were obtained from the Australian Red Cross Blood Service (ARCBS) after approval from the University of Melbourne Human Research Ethics Committee (1239046.2). Healthy human lung explant tissue was obtained via the Alfred Lung Biobank program and ARCBS from organs not suitable for donation after approval from the Blood Service HREC (2014#14) and University of Melbourne Human Research Ethics Committee (1545566.1).

**Compounds, immunogens and tetramers**. 5-OP-RU was prepared as described previously[18]. CpG1688 (sequence: T*C*C*A*T*G*A*C*G*T*T*C*C*T*-G*A*T*G*C*T (*phosphorothioate linkage) nonmethylated cytosine-guanosine oligonucleotide was purchased from Geneworks (Thebarton, Australia). Murine and human MR1 and β2-Microglobulin genes were expressed in *E. coli* inclusion bodies, refolded and purified as described previously[55]. MR1-5-OP-RU and MR1-6-FP tetramers were generated as described previously[4] and used in 1:200 dilution.

**Bacterial strains**. Cultures of *Legionella pneumophila* JR32 and *Legionella longbeachae* NSW150 were grown at 37 °C in buffered yeast extract (BYE) broth supplemented with 30–50 μg ml$^{-1}$ streptomycin for 16 h to log-phase (OD$_{600}$ 0.2–0.6) with shaking at 180 rpm. For the infecting inoculum, bacteria were re-inoculated in prewarmed medium for a further 2–4 h culture (OD$_{600}$ 0.2–0.6) with the estimation that 1 OD$_{600}$ = $5 \times 10^8$ ml$^{-1}$, sufficient bacteria were washed and diluted in PBS with 2% BYE for i.n. delivery to mice. A sample of inoculum was plated onto BYCE with streptomycin for verification of bacterial concentration by counting colony-forming units.

For infection of adoptive transfer donor mice with S. Typhimurium BRD509 cultures were prepared as previously described[13].

**Antibodies and flow cytometry**. Antibodies against murine CD4 (1:200 GK1.5, APCCy7), CD19 (1:200, 1D3, PerCP-Cy 5.5), CD45.2 (1:200, 104, FITC), IFN-γ (1:400, XMG1.2, PECy7), Ly6G (1:200, IA8, PECy7), TCRβ (1:200, H57-597, APC or PE), TNF (1:200, MP6-XT22, PE), GM-CSF (1:200, MP1-22E9, PE) and IL-17A (1:400, TC11-18H10, PE) were purchased from BD (Franklin Lakes, NJ). Antibodies against CD8a (1:800, 53-6.7, PE), PLZF (1:200, Mags.21F7, PE), RORγt (1:200, B2D, APC), T-bet (1:200, 4B10, PECy7) and MHCII (1:400, M5/114.15.2, AF700) were purchased from eBioscience (San Diego, CA). Antibodies against CD19 (1:200, 6D5, BV510), F4/80 (1:200, BM8, APC), CD11b (1:200, M1/70, FITC), CD11c (1:400, N418, BV786), CD31 (1:200, PCAM, MEC13.3, PerCPCy5.5), CD62L (1:200, Mel-14, FITC), CD64 (1:200, X54-5/71, BV711),

CD146 (1:200, ME-9F1, PerCPCy5.5) and CD326 (1:200, G8.8, EpCAM, APCCy7) were purchased from Biolegend (San Diego, CA). Blocking Ab (20 μg ml⁻¹, 26.5, 8F2.F9) and isotype controls (20 μg ml⁻¹, 3E12, 8A5) were prepared in-house. To block non-specific staining, cells were incubated with MR1-6-FP tetramer (nil conjugate, 1:100) and anti-Fc receptor (1:200, 2.4G2) for 15 min at room temperature and then incubated at room temperature with Ab/tetramer cocktails in PBS/2% foetal calf serum. 7AAD (5 μl per sample) was added for the last 10 min.

Antibodies against human CD3 (1:100, UCHT1, PE-AlexaFluor594), TCRVα7.2 (1:50, 3C10, APC), CD161 (1:100, HP-3G10, PECy7), TNF (1:50, Mab11, Pacific Blue) and viability dye (1:100, Zombie Yellow) were purchased from Biolegend. Antibodies against IFN-γ (1:50, 25725.11, FITC) and CD69 (1:50, FN50, PE) were purchased from BD, and anti-CD3 (1:100, UCHT1, APC) from eBioscience.

Cells were fixed with 1% paraformaldehyde prior to analysis on LSRII or LSR Fortessa or Canto II (BD Biosciences) flow cytometers. For intracellular cytokine staining Golgi plug (BD Biosciences) was used during all processing steps. Cells stimulated with PMA (phorbol 12-myristate 13-acetate)/ionomycin (20 ng ml⁻¹ and 1 μg ml⁻¹, respectively) for 3 h at 37 °C were included as positive controls. Surface staining was performed at 37 °C, and cells were stained for intracellular cytokines using the BD Fixation/Permeabilization Kit (BD Biosciences) or transcription factors using the transcription buffer staining set (eBioscience) according to the manufacturers' instructions. Flow cytometric data analysis was performed with FlowJo10 software (Ashland, OR).

**BrdU incorporation assay**. The assay was performed with the commercial BrdU In-Situ Kit (BD Biosciences) according to the manufacturer's instruction. Briefly, mice were infected at different time points for synchronous assays on a single day. BrdU (2 mg in 200 μl per mouse) was injected i.p. After 2 h, mice were killed for single-cell preparation from lungs and draining lymph nodes. Cells were stained with a cocktail of antibodies and MR1-5-OP-RU tetramers first, then fixed and permeabilised for intranuclear DNA staining with anti-BrdU antibody and other antibodies, as detailed above.

**Data availability**. Data supporting the findings of this study are available within the paper and its supplementary information files, or are available from the authors upon request.

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

## Acknowledgements

This research was supported by the National Health and Medical Research Council of Australia (NHMRC) Program Grants 1113293, 1016629 and 606788, a Project Grant 1120467, an ARC Centre of Excellence grant CE140100011, as well as a Merieux Research Grant. T.S.C.H. was supported by a Wellcome Trust Postdoctoral Research Fellowship (104553/z/14/z). A.J.C. was supported by an ARC Future Fellowship. S.B.G.E. was supported by an ARC DECRA Fellowship. D.I.G. is supported by an NHMRC Senior Principal Research Fellowship (1117766). J.R. was supported by an Australian ARC Laureate Fellowship. D.P.F. was supported by an NHMRC Senior Principal Research Fellowship. H.W. is supported by a Melbourne International Engagement Award (University of Melbourne). C.D.'S. is supported by a Melbourne International Research Scholarship and a Melbourne International Fee Remission Scholarship (University of Melbourne). We thank Dr Wei-Jen Chua and Prof Ted Hansen for their kind provision of 8F2.F9 and 26.5 mAbs, and Professor David Jackson for Pam2Cys. We are grateful to Professor Francis Carbone for critical review of the manuscript.

## Author contributions

H.W., C.D.'S., T.S.C.H., X.Y.L., L.K., T.J.P., S.B.G.E., B.S.M., Z.C., A.W.S. and M.S. performed the experiments and analysed the data. Z.C., T.S.C.H., H.W., J.M., A.J.C. and R.A.S. designed the experiments and managed the study. N.W., D.P.F., Y.I., J.M.G., G.P. W., L.K.-N., J.R., L.L., J.Y.W.M. and D.I.G. provided essential reagents and intellectual input. T.S.C.H., A.J.C., Z.C. and J.M. conceived the work and wrote the manuscript which was revised and approved by all authors.

## Additional information

**Competing interests:** Z.C., S.B.G.E., L.K-N., D.P.F., L.L., J.Y.W.M., J.R., J.McC. and A.J.C. declare that they are inventors on patents describing MR1 tetramers and MR1 ligands. The remaining authors declare no competing interests.

