## [Peer Review File · Nature Communications]

Reviewers' comments:

Reviewer #1 (Remarks to the Author):

The paper by Wang et al. entitled "Legionella protection and vaccination mediated by MAIT cells" presents data from mouse models indicating that MAIT cells play an important role in defense from pulmonary Legionella infection. Their results indicate that the protective effect of MAIT cells in this system is dependent on MR1, and mediated via IFN γ . The effect is enhanced in the absence of other T cells and NK cells. Finally, using MR1-presented antigen in conjunction with a TLR2 ligand, MAIT cells in lung can be boosted to provide an enhanced protection suggestive of a vaccine-like effect. This study is an important component in the ongoing work by many groups to demonstrate the importance of MAIT cells in the immune defense against infections. The paper is for the most part well written and likely to be of high interest to the field. Nevertheless is still suffers from a few shortcomings.

Major points:

Figure 1 is in vitro data on human MAIT cell responses to Legionella species, whereas the rest of the manuscript is based on mouse models. As far as this reviewer can see there is no demonstration of mouse MAIT cell responses to Legionella in vitro. It would be nice to add such data to Figure 1. Also, the rationale behind Figure 1D is unclear.

The authors nicely show that mouse pulmonary MAIT cells expand in vivo in response to Legionella infection. An important point is if this represents proliferation in the lung of a resident population, or recruitment from circulation or other sites. Is the response systemic or local? If recruitment, which factors govern this?

Similarly, the cytokine signature of lung MAIT cells changes over the course of infection (Figure 3). Is this pattern local or systemic?

In Figure 5, adoptive transfer of MAIT cells rescues Rag2^{-/-}gc^{-/-} mice from lethal pulmonary Legionella infection. After transfer the authors do perform CD8 and CD4 depletion. This is a bit odd, as CD8 and CD4 can be expressed on other cell types in vivo apart from adaptive T cells (incl MAIT cells) and this may skew results. At least this needs to be controlled for.

Data in Figure 6 seems to indicate that IFN γ is important for the protective effect of MAIT cells. In Figure 6B there is only three data points for IFN γ , is this a mistake? If not, more data points would be needed.

The authors speculate that IFN γ from MAIT cells mediate the protective effect by enhancing the activity of macrophages and neutrophils. This is indeed a strong possibility, and it would be nice if the authors could extend the data set with such data.

Minor points:

TNF α is nowadays known as TNF.

The wording of the title is a bit odd: "...protection and vaccination mediated...". Perhaps rephrase?

Reviewer #2 (Remarks to the Author):

In this manuscript Wang and colleagues report MR1-dependent activation of human and murine MAIT cells during Legionella infection and suggest that MAIT cells may exert a protective role against this bacterium, particularly in immune-compromised animals. Furthermore, their results indicate a potential role for vaccination to enhance MAIT cell immunity.

Specific comments:

- 1). Figure 1D: in the text it is stated that MAIT are found in the proximity of Legionella bacilli (line 131). I would argue that the only cell that is TCR Va7.2+, CD3+ and IL18R α + is not in proximity of the Legionella dot (yellow arrow). Furthermore, the detailed method for this experiment is missing. What is the blue Legionella staining detecting?
- 2). Figure 2A: it is not clear why the sample is from an animal that has been rechallenged. If so, one would expect a higher frequency of tetramer+ cells. The specificity controls with MR1 deficient mice and naïve WT mice are missing, and I think it is essential to show them as to my knowledge this is the first paper using MR1 tetramers in situ. Indeed the staining has high background, with several tetramer bright cells that are TCR negative, although this could also be due to TCR downregulation.
- 3). Supplementary Fig 2: There is no difference in the histological score between WT and MR1 deficient mice (panel C). Is there any difference in the bacterial load in each cell subset between WT and MR1 deficient mice (panel D)?
- 4). Figure 2C: The authors show expansion of MAIT cells during infection. Is it due to recruitment from the blood or to proliferation in situ? BrdU uptake and Ki67 staining should be performed.
- 5). Figure 3: The numbers of events acquired in panel A for ex vivo INF γ and GM-CSF is too low to unambiguously calculate the percentages (panel A, top). Why in naïve mice are 20% of MAIT cells already IL17A+, 1% INF γ + and 7% GM-CSF+ (panel B)? Is this the case also for conventional T cells incubated 4 hours with brefeldin A?
- 6). Figure 4: Interestingly, there is no difference in bacterial CFU at the peak of infection, rather from day 10 onwards, although in panel B I am unsure about the statistical significance of the data at day 14, as only 2 mice have low CFU. The comparison between Figures 4A, B and supplemental Fig S4 (line 216) is not appropriate as it is not the same experiment and the controls of mice treated with CpG without 5ARU is not included.

It is not clear why in panels D and E the authors have used Pam3Cy while in Fig S4 CpG. A naïve group is also missing in Fig 4E, and it is essential to control for the effect of Pam3Cys. As there is no difference in animal survival, the main criticism of this experiment is that there is no investigation on the effect of the marginally delayed bacterial clearance on initiation of adaptive immunity.
- 7). Figure 5: the authors show that adoptively transferred MAIT cells rescue Rag deficient mice from fatal Legionella infection. How specific is this effect for MAIT cells? The authors should compare MAIT with NK and conventional T cells.
- 8). Figure 6: In the adoptive transfer model, MAIT protection is MR1 and IFN- γ dependent. However, in the experiments shown in Fig 2 and S2 MAIT hardly make IFN γ , whether tested at day 7 or 100 post infection, despite Tbet expression. How do the authors explain this discrepancy? The cytokine secreting capacity of expanded MAIT cells in Rag deficient mice should be characterised.
- 9). The putative effect of MAIT-derived IFN γ on neutrophil and macrophage bactericidal activity

(line 325, discussion) is testable and the experiments should be included in this manuscript to dissect the molecular mechanism of MAIT cell protection.

Minor comments:

References 13, 26 and 33 are incomplete.

Reviewer #3 (Remarks to the Author):

In this manuscript, Wang et al. set out to examine the role of MAIT cells in host defense against *Legionella longbeachae* infection. They show that there is an expansion in the numbers of MAIT cells during pulmonary *L. longbeachae* infection, and that these cells persist over time. MAIT cells appear to have a very slight role in bacterial clearance at late timepoints, when the mice are largely already clearing infection. They show that preexpanding the MAIT population in mice by treating them with Pam2Cys and 5-OP-RU results in slight but accelerated bacterial clearance. To address the possibility that there is redundancy among MAIT cells and other lymphocyte populations, the authors examine the role of MAIT cells in Rag2^{-/-}gc^{-/-} mice. This is the most convincing data in the manuscript, as it shows that adoptive transfer of expanded MAIT cells from *Salmonella*-infected mice into Rag2^{-/-}gc^{-/-} mice protects from lethal infection. They further show that this protection requires IFN γ expression and to a lesser extent GM-CSF, but not TNF, IL-17, perforin, or Granzymes A and B. Overall, their data suggest that MAIT cells have a protective role during pulmonary *Legionella longbeachae* infection that is more evident in the absence of other lymphocytes. However, there are several critical issues that need to be addressed to strengthen their claims.

Major comments:

1. There are no statistics in figures 1A-C, making it difficult to draw any conclusions. In addition, it appears that the data shown in 1B-C are with cells from just one human donor. These experiments should be repeated independently with cells from at least two additional human donors to determine whether these are typical responses, or unique to this single donor.
2. For figure 1D, uninfected lung tissue should be shown in parallel.
3. For figure 2A and S2A, histology from uninfected control mice should be shown.
4. For figure 3A and S3A, the staining for some of the cytokines, such as GM-CSF, seems unconvincing. In general, the authors should include isotype control antibodies as controls for each of the cytokine stains. For figure 3B, the authors should also show total cell numbers to determine whether there are significant differences in numbers of cytokine producing MAIT cells.
5. In figures 3C and D, the authors should show isotype antibody control staining for T-bet and ROR γ t stains. In addition, the authors should show representative graphs with percentages and total cell numbers plotted for each of multiple mice, rather than show pie charts.
6. Did the authors measure cytokines in the BAL of infected Rag2^{-/-}gc^{-/-} plus or minus MAIT cells? As their findings suggest that IFN γ produced by MAIT cells is the key protective factor, it would have been nice if they could corroborate this by measuring IFN γ levels.

Minor comments:

1. Figure 4D and S2C should have statistics added.

2. Please switch order of Figure 4E and F, as 4F is described before 4E in manuscript text
3. Please specify in your title and abstract that you are using *Legionella longbeachae*, as all the in vivo experiments use *L. longbeachae*, and not *L. pneumophila*.
4. The authors should cite the Massis et al. paper, as they showed that IFN γ is important for control for *L. longbeachae* (*The Journal of Infectious Diseases*, Volume 215, Issue 3, 1 February 2017, Pages 440–451).

Reviewer #1:

Major points:

Figure 1 is *in vitro* data on human MAIT cell responses to Legionella species, whereas the rest of the manuscript is based on mouse models. As far as this reviewer can see there is no demonstration of mouse MAIT cell responses to Legionella in vitro. It would be nice to add such data to Figure 1.

Response: We believe that *in vivo* mouse data as presented in the manuscript best reflect the physiological responses of MAIT cells to infection. However, we agree that *in vitro* experiments would provide a nice link to the human data. Therefore, we have now performed additional experiments, which show *in vitro* activation by *Legionella longbeachae* lysates, as well as synthetic 5-OP-RU antigen control of mouse MAIT cells from $V\alpha 19iC\alpha^{-/-}MR1^{+}.Ly5.1$ mice, which are transgenic for the MAIT TCR.

These data are now shown in Figure 2A, and described in the associated text on page 7 line 138.

Also, the rationale behind Figure 1D is unclear.

Response: The purpose of Figure 1D and 2A (now 2B) is to provide a visual demonstration of MAIT cells, in their physiological context within the pulmonary architecture, and also to highlight the similarities of the human and murine biology. However, in light of the reviewer's comments, we have replaced Fig 1D with a more informative and relevant series of images. We contrast infected with uninfected sections and we now include nuclear counter-staining. Furthermore, in this new image we show both free and intracellular *Legionella* bacteria, the closest of which are within 35 micrometers of a MAIT cell, and some others within 55 micrometers of a heavily infected phagocytic cell (bright magenta). The revised figure legend now explicitly states that the magenta staining for *Legionella* is a polyclonal rabbit-anti-*legionella* antibody (page 41, line 884). We have also inserted the following additional text to illuminate the rationale (page 8, line 150):

“To visualise MAIT cells within their physiological context in situ we infected healthy human lung tissue ex vivo with L. longbeachae and observed CD3⁺TCRV α 7.2⁺ MAIT cells within the lung parenchyma in the proximity of free and phagocytosed Legionella bacilli 24 hours post-infection using immunofluorescence microscopy (Figure 1D).”

The authors nicely show that mouse pulmonary MAIT cells expand in vivo in response to Legionella infection. An important point is if this represents proliferation in the lung of a resident population, or recruitment from circulation or other sites. Is the response systemic or local? If recruitment, which factors govern this?

Response: The reviewer has raised an important question. For instance, in human studies, a decrease in MAIT cell numbers has been observed in the blood of patients with infections, including tuberculosis, and it has been hypothesised that MAIT cells traffic from the blood to the infected sites. The question of whether MAIT cells proliferate locally in tissues, in draining lymph nodes, or traffic from other sites is an important one in the field. To address this key question we have now employed a BrdU incorporation assay. This assay allows a snapshot of DNA synthesis, which reflects cell proliferation.

Mice were infected with 10^4 *L. Longbeachae* as described in the methods. At day 3 and 5 post infection mice were injected with BrdU. 2 hours later, mice were killed and BrdU positive (dividing) MAIT cells were enumerated. We now show representative FACS plots and percentage of BrdU⁺ MAIT and conventional T cells from lung and draining LN, in Figure 3 panels F, G.

In naïve mice, few BrdU⁺ MAIT cells were detected, reflecting minimal proliferation. At day 3 post-infection, a small BrdU⁺ MAIT cell population could be detected. By day 5 post-infection a significant proportion of BrdU⁺ MAIT cells could be detected in both lung and dLN (26% and 48% respectively).

Importantly, and unlike conventional T cells, a greater proportion of MAIT cells was dividing in the lung at day 3 post infection (lung 4.8% vs draining LN 2.4%) while this was reversed at day 5 (lung 26% vs LN 48%). This strongly suggests earlier activation and proliferation of MAIT cells locally in the lung tissue compared with the dLN. This conclusion is consistent with our previous work¹ in which MAIT cell activation was detected as early as 2 hours upon Ag encounter, indicating lung MAIT cells can be activated in local tissue without being primed via draining LN. The kinetics of conventional T cells in these experiments were consistent with a number of previously published studies^{2, 3, 4, 5}.

Consistent with our previous observation (*Chen et al. Mucosal Immunology, 2017*), we did not observe any reduction in MAIT cell percentage (of T cells) in the blood or other organs (data not shown). Furthermore, the total number of accumulated MAIT cells during infection in the lung is more than the estimated total number of MAIT cells in a whole naïve mouse ($<10^5$ /mouse: $<2 \times 10^4$ lung MAIT+ $<10^4$ liver MAIT+ $<4 \times 10^4$ Spleen MAIT+ $<10^4$ Kidney MAIT+ $<10^4$ Blood MAIT. Unpublished data), arguing mathematically against the notion that the MAIT cell accumulation in the lung was mainly by recruitment from other tissues.

Taken together, our data demonstrate that MAIT cells proliferate locally in both the draining LN and lung tissue and the accumulation during *Legionella* infection was mainly through local proliferation. We consider that recruitment from other tissues does not contribute significantly. This finding is also consistent with the previously published memory effector MAIT phenotypes⁶.

Accordingly, in addition to the new panels F and G in Figure 3, and updated figure legend, we have added two new paragraphs in the main text:

Page 10, line 200

A number of clinical studies have reported decreases in peripheral blood MAIT cell frequencies associated with pulmonary infections^{7, 8, 9, 10}, potentially attributable to recruitment from the blood. To address the question as to the source of MAIT cells at infection sites we used a BrdU incorporation assay, which enables quantitation of DNA synthesis, reflecting cell proliferation. As expected only a few BrdU⁺ MAIT cells (about 1%) and conventional T cells (about 1.3%) were enumerated in naïve mice (Figure 3F, far left, G) from both the lung and the draining mediastinal lymph nodes (dLN). These low percentages represent the minimum background staining, reflective of technical background artifact and possibly basal rates of self-renewal of pre-existing tissue resident memory (TRM) cells¹¹. As early as 3 DPI, an increased proportion of BrdU⁺ MAIT cells could be detected (lung, 4.8% and dLN, 2.4%) (Figure 3F, G). On day 5 post-infection, we detected a significant proportion of BrdU⁺ MAIT cells in both lung (26%) and dLN (48%) (Figure 3F, G). Interestingly the higher proportions of BrdU⁺ MAIT cells detected in the lungs compared with the dLN at 3 DPI was reversed at 5 DPI (Figure 3F, G). An equivalent change in these ratios was not observed among conventional T cells. This could indicate earlier activation and proliferation of MAIT cells locally in the lung tissue than in the dLN. Indeed this is consistent with our previous work in which MAIT cell activation was detected as early as 2 hours after

antigen encounter, indicating lung MAIT cells could be activated without being primed in the draining LN¹. Taken together, these data suggest that MAIT cells proliferate locally – both in the tissue and the dLN – upon infection. Furthermore they suggest pulmonary MAIT cell proliferation commences earlier than in the dLN, consistent with the notion that conventional T cell activation in the dLN requires additional time for the antigen-loaded dendritic cells to migrate to the dLN. The kinetics of conventional T cells were consistent with previous reports^{2, 3, 4, 5}. It is well recognised that naïve T cells need 3-4 days before they are observed proliferating in the dLN and are observed a day later at the site of inflammation².

Page 19, line 396

*The BrdU incorporation data demonstrated MAIT cell accumulation at the site of infection is largely due to local activation and proliferation. In several human studies^{7, 8, 10}, a decrease in MAIT cell numbers has been observed in the blood of patients with infections, including tuberculosis⁹, and it has been hypothesised that MAIT cells traffic from the blood to the infected sites. We did not observe any fall in peripheral blood frequencies during *L. longbeachae* infection. Rather, there was an increase of MAIT cells in blood and other organs (data not shown), consistent with our previous studies with *Salmonella* infection or vaccination^{1, 12}. It is still possible that in some infections inflammation-driven non-specific activation of MAIT cells^{8, 10, 13} could lead to MAIT cell exhaustion and premature apoptosis. However, this speculation remains to be tested experimentally in the future.*

Similarly, the cytokine signature of lung MAIT cells changes over the course of infection (Figure 3). Is this pattern local or systemic?

Response: We believe the local inflammatory cytokine milieu can drive changes in MAIT cell cytokine signatures, as for many conventional T cells¹⁴. As *Legionella* is largely a lung pathogen and does not disseminate widely if the mouse recovers from infection, we speculate that these changes in cytokine signature are initiated^{1, 12} at the local level. Our recent work^{1, 12} has shown that MAIT cells massively expand locally during lung infection, bearing the same cytokine signature, and subsequently re-distributing to other tissues and organs where some plasticity is observed, however unravelling the overall degree of functional plasticity of MAIT cells is beyond the scope of the current manuscript.

In Figure 5, adoptive transfer of MAIT cells rescues Rag2^{-/-}γc^{-/-} mice from lethal pulmonary Legionella infection. After transfer the authors do perform CD8 and CD4 depletion. This is a bit odd, as CD8 and CD4 can be expressed on other cell types in vivo apart from adaptive T cells (incl MAIT cells) and this may skew results. At least this needs to be controlled for.

Response: From our extensive experience with MAIT cell transfer to RAG2^{-/-}γc^{-/-} mice, we have observed that transferred conventional T cells will expand aggressively, such that contamination of enriched MAIT cell preparations with less than 1% conventional T cells will result in 8-10% conventional T cells after 14 days (Figure S6B, left panel), thus compromising our interpretation of results. Thus, we chose to deplete contaminating T cells using antibodies to CD4 and CD8, leaving CD4/8 double negative MAIT cells intact (Figure S6B). Indeed, with the low dose of antibodies used (2x 0.1mg each/mouse), we observed that the transferred MAIT cell CD4/8 composition profile was largely unchanged, while the contaminating conventional T cells were prevented from expanding (Figure S6C) (possibly due to differences in CD4/CD8 expression levels between MAIT cells and conventional T cells or lack of depletion of MAIT cells once resident in tissues such as the lung).

To address the concern that antibody depletion may affect other cell types we have infected RAG2^{-/-}γC^{-/-} mice with or without CD4/8 Ab depletion. In these experiments, the result was very similar with or without depletion both in terms of the survival curve and weight changes. Thus, the injection of low dose anti-CD4/8 Ab did not exert a noticeable effect on other cell types and the overall outcomes(Figure 6S6D, E).

We now show this new data in Supplementary Figure 4, and have made the following additions to the text (Page 15 line 324):

Although anti-CD4 and anti-CD8 mAbs were administered to deplete conventional CD4⁺ and CD8⁺ T cells, this exerted only a minor impact on eventual MAIT cell frequencies (Supplementary Figure S6B, C). Proportions of CD4⁺, CD8⁺ and CD4⁻ MAIT cells remained largely unchanged (Chen et al¹ and Supplementary Figure S6C). Moreover this temporary administration of anti-CD4 and anti-CD8 mAbs had no impact on weight loss or survival kinetics of Legionella-infected Rag2^{-/-}γC^{-/-} mice (Supplementary Figure S6D, E).

Data in Figure 6 seems to indicate that IFNγ is important for the protective effect of MAIT cells. In Figure 6B there is only three data points for IFNγ, is this a mistake? If not, more data points would be needed.

Response: We apologise for not explaining this data clearly. There were a total of 21 IFNγ^{-/-} mice (now Figure 7A). Among these mice, only three survived to 23 days post infection, at which point all mice were killed for assessment of bacterial load (CFU counts). Thus, only three data points were shown in the CFU data set for IFNγ^{-/-} mice (now Figure 7B).

In further experiments, any surviving mice (over 23 days post infection) were kept for longer. Infected IFNγ^{-/-} mouse in these subsequent experiments all died before 35 days post infection, whereas WT MAIT transferred mice could survive past this point. This result further strengthens our conclusion that IFN-γ is the key cytokine for MAIT cells to control *Legionella*.

In light of the reviewer's comments and to ensure clarity for readers, we have amended the text and the legend for Figure 6. Specifically, we now state on page 17 line 355:

“By contrast protection was critically dependent on MAIT cell derived IFN-γ: when donor MAIT cells were deficient in IFN-γ survival was decreased (P<0.0001) and there was a 2.8 log-fold increase in bacterial burden (P<0.001) amongst the small proportion (3/21) of animals which survived till 23 DPI, at which point all such mice succumbed to Legionella infection.”

The authors speculate that IFNγ from MAIT cells mediate the protective effect by enhancing the activity of macrophages and neutrophils. This is indeed a strong possibility, and it would be nice if the authors could extend the data set with such data.

Response: IFNγ is known to mediate effects on the innate immune cells by a variety of mechanisms, and our findings suggest that MAIT cell-derived IFNγ mediates its effect through one of these previously defined pathways. We appreciate the reviewer's comments, but feel that assessment of the effect on the innate immune response to *Legionella* lies beyond the reach of

the current manuscript. We have added this point to our discussion, on page 20, line 425, where we now state:

*“Furthermore T cell-derived IFN- γ has been shown to stimulate the bactericidal activity of monocyte derived cells in murine *L. pneumophila* infection¹⁵”.*

We have performed preliminary experiments to address the effect of MAIT-cell-derived IFN γ on the innate immune system by measuring reactive oxygen species (ROS) production by neutrophil and macrophages from naïve and infected RAG $^{-/-}$ γ C $^{-/-}$ mice that had previously received transferred wt vs IFN γ $^{-/-}$ MAIT cells. Our data show a trend towards higher ROS production from both cell types in the presence of MAIT cells that were capable of producing IFN- γ than those that were deficient in IFN- γ production. Thus, this may be one mechanism by which MAIT cells mediate indirect innate immune effects. However, we are reluctant to add these preliminary data to the manuscript since we believe a full characterization this would require significant new studies and new techniques, beyond the scope of the current manuscript.

Figure legend:

MAIT cell (wt vs IFN γ $^{-/-}$) transferred RAG $^{-/-}$ γ C $^{-/-}$ mice were prepared as described in the manuscript. After two weeks resting, mice were infected with 5×10^3 CFU *L. longbeachae*, or left uninfected. At 3 days post infection (dpi), mice were killed and lung cells prepared. ROS staining was performed with a commercial kit (DCFDA Cellular ROS Detection Assay Kit from abcam) by following manufacturer’s instruction. ROS production was determined by measuring the intracellular staining of ROS by flow cytometry. Mean flow intensity from each sample was normalised against the average MFI from wt-MAIT transferred, uninfected RAG $^{-/-}$ γ C $^{-/-}$ mice, displayed aside Y-axis as “Relative ROS production”.

Minor points:

TNFalpha is nowadays known as TNF.

Response: We have now made this change throughout the manuscript and associated figures.

The wording of the title is a bit odd: “...protection and vaccination mediated....”. Perhaps rephrase?

Response: In light of the reviewer’s comments, and to specify the use of *Legionella longbeachae* as requested, we have changed the “Mucosal Associated Invariant T (MAIT) cells protect against pulmonary *Legionella longbeachae* infection”

Reviewer #2:

1). Figure 1D: in the text it is stated that MAIT are found in the proximity of Legionella bacilli (line 131). I would argue that the only cell that is TCR V α 7.2+, CD3+ and IL18R α + is not in proximity of the Legionella dot (yellow arrow). Furthermore, the detailed method for this experiment is missing. What is the blue Legionella staining detecting?

Response: In light of the reviewer's comments, we have replaced Figure 1D with a more informative and relevant series of images. We contrast infected with uninfected sections and we now include nuclear counter-staining. Furthermore, in this new image we show both free and intracellular legionella bacteria, the closest within 35 micrometers of a MAIT cell, and others within 55 micrometers of a heavily infected phagocytic cell. The revised figure legend now explicitly states that the magenta staining for legionella is a polyclonal rabbit-anti-legionella antibody (page 41, line 884).

As requested, the detailed method has been included with the insertion of the following additional text (Page 26, line 553):

“Following rehydration, the unfixed cryopreserved 8 μ m sections of human lung were blocked with serum-free block and stained with purified anti-V α 7.2 TCR mAb clone 3C10 (Biolegend), rat-anti-human CD3 mAb (BioRad) and polyclonal rabbit anti-Legionella antibody. After 1 hour at room temperature, sections were washed and stained with a cocktail of secondary antibodies containing goat anti-mouse Alexa Fluor 568, goat anti-rat Alexa Fluor 488 and goat anti-rabbit Alexa Fluor 647, all from Life Technologies. Nuclei were counterstained with Hoechst 33342 (Life Technologies).”

2). Figure 2A: it is not clear why the sample is from an animal that has been rechallenged. If so, one would expect a higher frequency of tetramer+ cells. The specificity controls with MR1 deficient mice and naïve WT mice are missing, and I think it is essential to show them as to my knowledge this is the first paper using MR1 tetramers in situ. Indeed the staining has high background, with several tetramer bright cells that are TCR negative, although this could also be due to TCR downregulation.

Response: Total numbers of MAIT cells are low in mice, especially in naïve mice and in early infection. Nonetheless, “MAIT cells” were NOT present in MAIT deficient MR1 $^{-/-}$ mice, nor in mice that had not previously been infected. As requested we now show the specificity controls in a fully revised Figure 2B, which now includes both MR1 $^{-/-}$ and C57BL6 mice. We also now include the MR1-6-FP tetramer control, which controls for background staining with the MR1 tetramer, and demonstrates that MR1-5-OP-RU tetramer positive cells are clearly distinguishable from background, especially when interpreted in the context of co staining with TCR β .

Therefore, in light of the reviewer's comment we now include a much-expanded Figure 2, which includes multiple controls, to give the readers confidence in our findings, and also includes both naïve and infected (3DPI) mice with negative control MR1-6FP tetramer staining. Furthermore, in light of the reviewer's comment we have now added the following additional text (page 7, line 129):

*“To visualise murine MAIT cells within their physiological context in situ we infected wild-type C57BL/6 (MAIT sufficient) and MR1 $^{-/-}$ (MAIT deficient) mice with *L. longbeachae* and observed MAIT cells (Figure 2B) within the lung parenchyma. Despite high background staining, we believe the MR1-5-OP-RU tetramer $^{+}$, TCR β $^{+}$ images (highlighted in row 3) are MAIT cells, constituting the first demonstration of immunofluorescent staining of MAIT cells with murine MR1.*

Importantly similar images were not observed in negative controls using the same tissue stained with non-specific MR1-6-FP tetramer (row 4) or infected MR1^{-/-} mice (row 2). The low frequencies of MAIT cells in naïve mice precluded similar co-stained cell images (row 1).”

3). Supplementary Fig 2: There is no difference in the histological score between WT and MR1 deficient mice (panel C). Is there any difference in the bacterial load in each cell subset between WT and MR1 deficient mice (panel D)?

Response: As no difference was observed either in histology (Fig S3E) or in the total bacterial load (Fig 5A), we believe it is unlikely that any difference would be seen in the bacterial load in each corresponding cell subsets between WT and MR1^{-/-} mice, considering big variations among individual mice.

4). Figure 2C: The authors show expansion of MAIT cells during infection. Is it due to recruitment from the blood or to proliferation in situ? BrdU uptake and Ki67 staining should be performed.

Same question asked by reviewer 1

Response: The reviewer has raised an important question in the field. For instance, in human studies, a decrease in MAIT cell numbers has been observed in the blood of patients with infections, including tuberculosis, and it has been hypothesised that MAIT cells traffic from the blood to the infected sites. The question of whether MAIT cells proliferate locally in tissues, in draining lymph nodes, or traffic from other sites is an important one in the field. To address this key question we have now employed a BrdU incorporation assay. This assay allows a snapshot of DNA synthesis, which reflects cell proliferation.

Mice were infected with 10^4 *L. Longbeachae* as described in the methods. At day 3 and 5 post infection mice were injected with BrdU. 2 hours later, mice were killed and BrdU positive (dividing) MAIT cells were enumerated. We now show, in Figure 3 panels F, G, representative FACS plots and percentage of BrdU +ve MAIT and conventional T cells from lung and draining LN.

In naïve mice, few BrdU⁺ MAIT cells were detected, reflecting minimal proliferation. At day 3 post-infection, a small BrdU⁺ MAIT cell population could be detected. By day 5 post-infection a significant proportion of BrdU⁺ MAIT cells could be detected in both lung and dLN (26% and 48% respectively).

Importantly, and unlike conventional T cells, a greater proportion of MAIT cells was dividing in the lung at day 3 post infection (lung 4.6% vs draining LN 2.4%) while this was reversed at day 5 (lung 26% vs LN 48%). This strongly suggests earlier activation and proliferation of MAIT cells locally in the lung tissue than that in the dLN. This conclusion is indeed in line with our previous work¹ in which MAIT cell activation was detected as early as 2hours upon Ag encounter, indicating lung MAIT cells can be activated in local tissue without being primed via draining LN. The kinetics of conventional T cells were consistent with a number of previously published studies^{2, 3, 4, 5}.

Consistent with our previous observation (*Chen et al. Mucosal Immunology, 2017*), we did not observe any reduction in MAIT cell percentage (of T cells) in the blood or other organs (data not shown). Furthermore, the total number of accumulated MAIT cells during infection in the lung is more than the estimated total number of MAIT cells in a whole naïve mouse ($<10^5$ /mouse: $<2 \times 10^4$ lung MAIT+ $<10^4$ liver MAIT+ $<4 \times 10^4$ Spleen MAIT+ $<10^4$ Kidney MAIT+ $<10^4$ Blood MAIT.

Unpublished data), arguing mathematically against the notion that the MAIT cell accumulation in the lung was mainly by recruitment from other tissues.

Taken together, our data demonstrate that MAIT cells proliferate locally in both the draining LN and lung tissue and the accumulation during *Legionella* infection was mainly through local proliferation. We consider that recruitment from other tissues does not contribute significantly. This finding is also consistent with the previously published memory effector MAIT phenotypes⁶.

According, in addition to the new panels F and G in Figure 3, and updated figure legend, we have added two new paragraphs in the main text:

- Page 10, line 200-p11, line 226, a new paragraph.

A number of clinical studies have reported decreases in peripheral blood MAIT cell frequencies associated with pulmonary infections^{7, 8, 9, 10}, potentially attributable to recruitment from the blood. To address the question as to the source of MAIT cells at infection sites we used a BrdU incorporation assay, which enables quantitation of DNA synthesis, reflecting cell proliferation. As expected only a few BrdU⁺ MAIT cells (approximately 1%) and conventional T cells (about 1.3%) were enumerated in naïve mice (Figure 3F, far left, G) from both the lung and the draining mediastinal lymph nodes (dLN). These low percentages represent the minimum background staining, and basal rates of self-renewal of pre-existing tissue resident memory (TRM) cells¹¹. As early as 3 DPI, an increased proportion of BrdU⁺ MAIT cells could be detected (lung, 4.8% and dLN, 2.4%) (Figure 3F, G). On day 5 post-infection, we detected a significant proportion of BrdU⁺ MAIT cells in both lung (26%) and dLN (48%) (Figure 3F, G). Interestingly the higher proportions of BrdU⁺ MAIT cells detected in the lungs compared with the dLN at 3 DPI was reversed at 5 DPI (Figure 3F, G). An equivalent change in these ratios was not observed among conventional T cells. This could indicate earlier activation and proliferation of MAIT cells locally in the lung tissue than in the dLN. Indeed this is consistent with our previous work in which MAIT cell activation was detected as early as 2 hours after antigen encounter, indicating lung MAIT cells could be activated without being primed in the draining LN¹. Taken together, these data suggest that MAIT cells proliferate locally – both in the tissue and the dLN – upon infection. Furthermore they suggest pulmonary MAIT cell proliferation commences earlier than in the dLN, consistent with the notion that conventional T cell activation in the dLN requires additional time for the antigen-loaded dendritic cells to migrate to the dLN. The kinetics of conventional T cells were consistent with previous reports^{2, 3, 4, 5}. It is well recognised that naïve T cells need 3-4 days before they are observed proliferating in the dLN and are observed a day later at the site of inflammation².

- Page 19, line 388-p20, line 398, a new paragraph.

*The BrdU incorporation data demonstrated MAIT cell accumulation at the site of infection is largely due to local activation and proliferation. In several human studies^{7, 8, 10}, a decrease in MAIT cell numbers has been observed in the blood of patients with infections, including tuberculosis⁹, and it has been hypothesised that MAIT cells traffic from the blood to the infected sites. We did not observe any fall in peripheral blood frequencies during *L. longbeachae* infection. Rather, there was an increase of MAIT cells in blood and other organs (data not shown), consistent with our previous studies with *Salmonella* infection or vaccination^{1, 12}. It is still possible that in some infections inflammation-driven non-specific activation of MAIT cells^{8, 10, 13} could lead to MAIT cell exhaustion and premature apoptosis. However, this speculation remains to be tested experimentally in the future.*

5). Figure 3: The numbers of events acquired in panel A for ex vivo INF γ and GM-CSF is too low to unambiguously calculate the percentages (panel A, top). Why in naïve mice are 20% of MAIT

cells already IL17A+, 1% IFN γ + and 7% GM-CSF+ (panel B)? Is this the case also for conventional T cells incubated 4 hours with brefeldin A?

Response: We agree the dot plots shown in Figure 3A (now 4A) for *ex vivo* IFN γ and GM-CSF are visually too low to convince readers. As pseudocolour plots can support many more events than dot plots, we replaced the dot plots in Figure 3A (now 4A) with corresponding pseudocolour plots. Please note the two plots with GM-CSF staining have been replaced with better representative plots. To clarify, the numbers of gated (MAIT cell) events analysed are around 7000 (below). We believe allows clear visual assurance of percentages.

We believe the relatively high basal expression of cytokines by MAIT cells, as detected by ICS, may be due to a combination of two factors. Firstly, basal activation in the lungs may occur since this is not a sterile environment. Indeed, MAIT cell expansion in the periphery requires the presence of microflora (Germ-free mice have minimal MAIT cells). Secondly, many cytokines are pre-made and stored in cells, and would thus be detected by ICS staining methods. Since MAIT cells display a memory effector phenotype (CD62L^{LOW}, CD44^{HIGH})^{6,16}, they may have higher levels of pre-stored cytokines than naïve conventional T cells. To make this comparison directly, we now show additional data from conventional T cells in Figure 4B (and below for your convenience).

Accordingly, we have made changes in the manuscript text (Page 11, line 228):

*“To explore MAIT cell function we investigated the dynamics of their cytokine profile throughout infection. During acute *L. longbeachae* infection MAIT cells secreted interleukin IL-17A, IFN- γ , GM-CSF (Figure 4A, Supplementary Figure S4A, B) and TNF (data similar to IFN- γ , not shown). The percentage of IL-17A-expressing MAIT cells was high and increased slightly throughout the*

course of infection (naïve 22%, 7 DPI 27% and >100 DPI 30%; Figure 4B left panel), whilst IFN- γ secretion was proportionately less, but nevertheless was significantly higher during the acute infection than in uninfected cells or after resolution (each $P < 0.005$, Figure 4B middle panel). Conversely, the percentage of GM-CSF-expressing MAIT cells was lowest during acute infection and peaked after disease resolution ($P = 0.0004$ acute v resolution). Absolute numbers of IL-17A expressing MAIT cells increased 200-fold from baseline in acute infection and remained 27-fold increased even after resolution of infection (Figure 4C, left panel). The greatest differences observed were in IFN- γ -secreting MAIT cells, which increased 300-fold in acute infection, contracting during resolution to 12-fold above baseline (Figure 4C, middle panel). Numbers of GM-CSF-secreting MAIT cells were increased 89- and 41-fold during acute infection and resolution (Figure 4C, right panel). No changes were observed with isotype (supplementary Figure S5A, B).”

6). Figure 4: Interestingly, there is no difference in bacterial CFU at the peak of infection, rather from day 10 onwards, although in panel B I am unsure about the statistical significance of the data at day 14, as only 2 mice have low CFU.

Response: We agree that the difference in CFU at later time points is interesting. To address this, we state in our discussion, on page 13 line 267:

“In normal C57BL/6 mice we observed a significant difference in bacterial load, but not until days 10 and 14 post infection, immediately subsequent to the time of peak MAIT cell numbers at 7-9 DPI.”

We believe the reviewer must have overlooked the data points on X axis, which show clearance of bacteria in many C57BL/6 mice at day 14 (6 mice had bacterial counts below the limit of detection). To ensure clarity in the presentation of this data for the reader, we have changed the symbols from closed triangles to more visually friendly open circles for C57BL/6 mice and open squares for MR1^{-/-} mice (now Figure 5B). The dashed line indicating the limit of detection has been reduced in intensity to ensure it does not visually impact the data points.

The comparison between Figures 4A, B and supplemental Fig S4 (line 216) is not appropriate as it is not the same experiment and the controls of mice treated with CpG without 5ARU is not included.

Response: We agree with the reviewer’s comment on Figures 4A, B and supplemental Fig S4. “(compare Figures 4A, B to Supplementary Figure S4)” now is amended to “(Figure 5F)”.

We have also made the following changes for the this old Figure 4.

1. Figure 4 is now renumbered as Figure 5
2. Moved S4 to the main Figure 5 as 5E.
3. Moved and renumbered old Figure 4E, F to Figure 5F, G.

It is not clear why in panels D and E the authors have used Pam3Cy while in Fig S4 CpG. A naïve group is also missing in Fig 4E, and it is essential to control for the effect of Pam3Cys. As there is no difference in animal survival, the main criticism of this experiment is that there is no investigation on the effect of the marginally delayed bacterial clearance on initiation of adaptive immunity.

Response: We have previously found Pam2Cys (or Pam2lys) and CpG have similar effect as danger signals in combination with 5-OP-RU to initiate MAIT response¹.

We agree with the reviewer's comment that there was no survival difference between C57BL/6 vs MR1^{-/-} mice due the small incremental effect of MAIT cells. However we disagree that there was no investigation of the effect of the marginally delayed bacterial clearance on adaptive immunity. In Figure 3D, we have demonstrated the kinetics of conventional T cells were almost identical in C57BL/6 mice and MR1^{-/-} mice through the infection course, suggesting there is no significant effect from the marginally delayed bacterial clearance on initiation of adaptive immunity. We particularly compared the potential recruitment effect on CD4⁺ T cells as shown previously¹⁷. No significant difference in our *Legionella* infection setting in CD4⁺ T cells numbers was observed throughout the infection course between C57BL/6 and MR1^{-/-} mice. Furthermore, when CD4⁺ T cells are removed in the GK1.5 mice (Figure 5G), more prominent protection from MAIT cells was revealed. This result further diminishes the possibility that the augmented protection in C57BL/6 over MR1^{-/-} mice were solely from the presumed extra CD4⁺ T cells by additional moDCs *via* recruitment mediated by MAIT cells in *legionella* infection.

7). Figure 5: the authors show that adoptively transferred MAIT cells rescue Rag deficient mice from fatal Legionella infection. How specific is this effect for MAIT cells? The authors should compare MAIT with NK and conventional T cells.

Response: We understand with the reviewer's concern on the specificity of MAIT cells' protection and have attempted to address the contribution of MAIT cells by the administration of anti-MR1 blocking monoclonal antibody (shown in the renumbered Figure 6D and 6E). The experiment demonstrated that the protection that is provided by transferred MAIT cells was largely abolished by MR1-blocking Ab; this is reflected in the sharp drop in survival in these mice (Figure 6D) and significantly higher bacterial load in surviving mice (Figure 6E). One would appreciate that most antibody blockings are often leaky and never be totally complete.

For the question of NK cell contributions, we have previously performed the equivalent experiment with RAG2^{-/-} mice, which lack both T and B cells, but have intact NK cell immunity. Our results from these experiments showed that NK cells are capable of controlling *Legionella* infection. In the presence of NK cells, these knockout mice survived *Legionella* challenge and bacteria were eventually cleared by day14 (supplementary Figure S6H). Thus, we chose to examine the role of MAIT cells in the RAG2^{-/-}γC^{-/-} mice. Our logic is that by peeling away the layers of protective immunity, as may be reflective of immunocompromised humans, we could examine the capacity for MAIT cells to protect against *Legionella* infection. To address the reviewer's concerns and to clarify our logic in this section, we have made changes to the manuscript text. Specifically, we now state (on page 16 line 333):

“We also assessed Rag2^{-/-} mice, which lack T and B cells but have an intact NK population. These mice recover well from pulmonary Legionella infection (2x 10⁴ CFU) (Supplementary Figure S6H), indicating a role for NK cell immunity.”

With respect to conventional T cells, we have conducted experiments directly comparing the protection capacity of conventional T cells and MAIT cells, we showed conventional T cells can rescue RAG2^{-/-}γC^{-/-} mice from fatal *Legionella* infection. When equal numbers of CD4⁺, CD8⁺ or MAIT cells were adoptively transferred, CD4⁺ conventional T cells provided complete protection compared to similar outcomes for CD8⁺ conventional T cells or MAIT cells transferred mice, as

measured by survival rate and bacteria load.(Figures below, and also included in supplementary Figure S6F, G and below for your convenience).

Legend: Equal number (10^5) of CD4+, CD8+ or MAIT cells were transferred via tail vein injection. Residual contaminant cells were depleted by i.p injection twice in the first week after cell transfer, with anti-CD8, anti-CD4 and a combination anti-CD4 anti-CD8 (0.1mg each) separately to the corresponding T cell transferred RAG2^{-/-}γC^{-/-} mice. Two weeks later mice were then infected i.n with 10^4 CFU *L. longbeachae*. Mouse survival and bacterial loads were worked similarly as described in the manuscript (Figure 5).

Accordinging changes in the main text on page16, line 330:

“Direct comparison with conventional CD4⁺ or CD8⁺ T cells showed that MAIT cells could protect mice from lethal challenge similarly to CD8⁺ conventional T cells, and slightly less than CD4⁺ conventional T cells in terms of both survival kinetics and bacterial load (Supplementary Figure S6F, G).”

8). Figure 6: In the adoptive transfer model, MAIT protection is MR1 and IFN-γ dependent. However, in the experiments shown in Fig 2 and S2 MAIT hardly make IFN-γ, whether tested at day 7 or 100 post infection, despite Tbet expression. How do the authors explain this discrepancy? The cytokine secreting capacity of expanded MAIT cells in Rag deficient mice should be characterised.

Response: We agree with the comment that MAIT protection is MR1 and IFNγ dependent and the cytokine secreting capacity of expanded MAIT cells in RAG deficient mice should be characterised. However, we do not agree that the MAIT cells shown in Fig 2 and S2, MAIT hardly make IFNγ. The low expression level of IFNγ in MAIT cells is real and physiologically important, as shown by our data in Figure 5A and 5B.

As a consequence of the reviewers’ suggestion, we have now performed ICS on MAIT cells transferred to RAG^{-/-}γC^{-/-} mice. Representative FACS plots are shown in Figure 7C and the statistics were shown in and Figure 7D. The transferred MAIT cells displayed an increased level of every cytokine examined following *Legionella* challenge, compared to those in C57BL/6 mice. The key cytokine IFNγ was expressed at a significantly higher level, from similar basal level of 0.82% in uninfected RAG^{-/-}γC^{-/-} mice (Figure 7C, top middle panel) as in C57BL/6 mice (Figure 4B, middle graph) to 13.7% at d7-PI in RAG^{-/-}γC^{-/-} mice (Figure 7C lower middle panel). The increase is massive when comparing with the counterparts in C57BL/6 mice (from 1% to 1.7%, Figure 4B, middle graph). This result partially explained the more dramatic protection of MAIT cells in RAG^{-/-}γC^{-/-} mice than the more subtle delay of several days in bacterial clearance in MR1^{-/-} vs C57BL/6 mice. More importantly it also recapitulates the concept that MAIT cells do establish immune memory,

with which transferred MAIT cells (primed) are augmented in quality and quantity, conferring better and faster protection (Figure 5E, F).

Accordingly, we have made changes in the main text on page 17, line 362:

“Consistent with this protection, transferred MAIT cells **expressed relevant pro-inflammatory cytokines (IL-17A, IFN- γ , GM-CSF)** during *L. longbeachae* challenge. **Adoptively-transferred MAIT cells maintained an effector-memory cytokine profile in uninfected mice**, similar to their corresponding 100 DPI counterparts, with the exception of GM-CSF (compare Figure 4B with Figure 7D). Most strikingly, MAIT cells **increased IFN- γ expression significantly by >5 fold from 1.7% in C57BL/6 (Figure 4A, B) to 9.8% (Figure 7C, D), and GM-CSF expression increased 27 fold (3.3% to 9.1%) whilst IL-17 remained largely unchanged (27% to 30%).**”

9). The putative effect of MAIT-derived IFN γ on neutrophil and macrophage bactericidal activity (line 325, discussion) is testable and the experiments should be included in this manuscript to dissect the molecular mechanism of MAIT cell protection.

Response: We have performed preliminary experiments to address the effect of MAIT-cell-derived IFN γ on the innate immune system by measuring reactive oxygen species (ROS) production by neutrophil and macrophages from naïve and infected RAG^{-/-} γ C^{-/-} mice that had previously received transferred wt vs IFN γ ^{-/-} MAIT cells. Our data show a trend towards higher ROS production from both cell types in the presence of MAIT cells that were capable of producing IFN γ than those that were deficient in γ production. Thus, this may be one mechanism by which MAIT cells mediate indirect innate immune effects. However, we are reluctant to add these preliminary data to the manuscript since we believe a full characterization this would require significant new studies and new techniques, beyond the scope of the current manuscript.

Figure legend:

MAIT cell (wt vs IFN γ ^{-/-}) transferred RAG^{-/-} γ C^{-/-} mice were prepared as described in the manuscript. After two weeks resting, mice were infected with 5x10³ CFU *L. longbeachae*, or left uninfected. At 3 days post infection (dpi), mice were killed and lung cells prepared. ROS staining was performed with a commercial kit (DCFDA Cellular ROS Detection Assay Kit from abcam) by following manufacturer's instruction. ROS production was determined by measuring the intracellular staining of ROS by flow cytometry. Mean flow intensity from each sample was normalised against the average MFI from wt-MAIT transferred, uninfected RAG^{-/-} γ C^{-/-} mice, displayed as Y-axis as "Relative ROS production".

Minor comments:

References 13, 26 and 33 are incomplete.

We thank the reviewer for noticing this error in reference 13. We have now corrected this in the reference list. Note that references 26 and 33, like all the other references are complete according to the reference manager output specifications for this journal, but further adjustments could be made at copyediting stage if required.

Reviewer #3 (Remarks to the Author):

Major comments:

1. There are no statistics in figures 1A-C, making it difficult to draw any conclusions. In addition, it appears that the data shown in 1B-C are with cells from just one human donor. These experiments should be repeated independently with cells from at least two additional human donors to determine whether these are typical responses, or unique to this single donor.

Response: As requested, Figures 1A have now been revised by the addition of the relevant statistical tests and insertion of the following additional text in the figure legend (page 41, line 871):

“Experiment performed in triplicate wells on two separate occasions with similar results. Statistical tests: One way ANOVA and post hoc Dunnett’s comparing all columns with the first column (black). Unpaired T test (blue).”

The data presented in old figure 1B-C are in fact from three different human donors, tested on experiments performed on two separate days, and therefore we believe these are indeed typical responses.

To completely avoid any doubt, we also performed a new experiment with a further 6 donors’ PBMCs. New data are presented in a new figure S1A-B with individual data points displayed. These findings are entirely consonant with the previous dataset.

2. For figure 1D, uninfected lung tissue should be shown in parallel.

Response: As requested we now include both uninfected and infected lung tissue in parallel in a revised Figure 1D. In this new image we show both free and intracellular *legionella* bacteria, the closest within 35 micrometers of a MAIT cell, and within 55 micrometers of a heavily infected phagocytic cell.

3. For figure 2A and S2A, histology from uninfected control mice should be shown.

Response: As requested we now show histology from uninfected control mice in a fully revised Figure 2 and Figure S2 (now S3) and have updated the Figure legends accordingly.

4. For figure 3A and S3A, the staining for some of the cytokines, such as GM-CSF, seems unconvincing. In general, the authors should include isotype control antibodies as controls for each of the cytokine stains. For figure 3B, the authors should also show total cell numbers to determine whether there are significant differences in numbers of cytokine producing MAIT cells.

Response: We now show isotype controls for the cytokine staining (Figure S5A, B), and have updated the Figure legends accordingly. All isotype controls displayed basal level staining and no shift in these controls was observed in the PMA + Ionomycin stimulated samples.

In order to determine whether there were significant differences in numbers of cytokine-producing MAIT cells, we have now reanalyzed our data, and now show the populations of MAIT cells expressing individual-cytokine in Figure 4B and 4C (as percentage and absolute numbers). As expected, the population of cytokine producing MAIT cells follow the kinetics of total MAIT cells during the course of infection, peaking in number at day 7 post infection. The increase in cytokine producing MAIT cells was significant: 208-fold for IL-17⁺; 303-fold for IFN γ ⁺ and 89-fold for GM-CSF⁺ MAIT cells. Interestingly even after bacteria were cleared (>d100-PI), the cytokine positive MAIT cell populations remained significantly higher than in uninfected mice 27-fold for IL-17⁺; 12-fold for IFN γ ⁺ and 42-fold for GM-CSF⁺ MAIT cells.

Accordingly, we have made the following changes in the manuscript text (Page 11, line 238):

“Absolute numbers of IL-17A expressing MAIT cells increased 200-fold from baseline in acute infection and remained 27-fold increased even after resolution of infection (Figure 4C, left panel). The greatest differences observed were in IFN- γ -secreting MAIT cells, which increased 300-fold in acute infection, contracting during resolution to 12-fold above baseline (Figure 4C, middle panel). Numbers of GM-CSF-secreting MAIT cells were increased 89- and 41-fold during acute infection and resolution (Figure 4C, right panel). No changes were observed with isotype (supplementary Figure S5A, B).”

5. In figures 3C and D, the authors should show isotype antibody control staining for T-bet and ROR γ t stains. In addition, the authors should show representative graphs with percentages and total cell numbers plotted for each of multiple mice, rather than show pie charts.

Response: Transcription factor staining was validated using the appropriate isotype controls (mouse IgG1k for T-bet, rat IgG1k for ROR γ t, eBioscience). As requested, these plots are now shown in Supplementary Figure S5C.

As requested, we now show both representative plots and percentages and total cell numbers for individual mice in Supplementary Figure S4C and S4D, in addition to the pie charts shown in Figure 4E. To clarify, these pie charts were plotted from many mice, and we believe show a clear visual representation of the MAIT cell profile transition during the course of infection.

We have adjusted the text on page 12 line 247:

“In uninfected mice most (81 \pm 4%, mean \pm SD) MAIT cells expressed the orphan nuclear receptor, retinoic acid-related orphan receptor γ (ROR γ t) alone: a master regulator of Th17 cell differentiation (Figure 4D, E, Supplementary Figure S4C, D). Isotype controls are shown in supplementary Figure S5C. “

6. Did the authors measure cytokines in the BAL of infected Rag2^{-/-}gc^{-/-} plus or minus MAIT cells? As their findings suggest that IFN γ produced by MAIT cells is the key protective factor, it would have been nice if they could corroborate this by measuring IFN γ levels.

Response: As a consequence of the reviewer’s suggestion, we have now performed ICS on transferred MAIT cells to RAG^{-/-} γ C^{-/-} mice. Representative FACS plots are shown in Figure 7C and the statistics were shown in and Figure 7D. The transferred MAIT cells displayed an increased level for every cytokine examined upon *Legionella* challenge, compared to those in C57BL/6 mice. The key cytokine IFN γ was expressed at a significantly higher level, from similar basal level of 0.82% in uninfected RAG^{-/-} γ C^{-/-} mice (Figure 7C, top middle panel) as in C57BL/6 mice (Figure 4B, middle

graph) to 13.7% at d7-PI in RAG^{-/-}γC^{-/-} mice (Figure 7C lower middle panel). The increase is massive when comparing with the counterparts in C57BL/6 mice (from 1% to 1.7%, Figure 4B, middle graph). This result partially explained the more dramatic protection of MAIT cells to RAG^{-/-}γC^{-/-} mice than the more subtle delay of several days in bacterial clearance in MR1^{-/-} vs C57BL/6 mice. More importantly it also recapitulates the concept that MAIT cells do establish immune memory, with which transferred MAIT cells (primed) are augmented in quality and quantity, conferring better and faster protection (Figure 5E, F).

Accordingly we have made changes in the main text on page 17, line 362:

Consistent with this protection, transferred MAIT cells expressed relevant pro-inflammatory cytokines (IL-17A, IFN-γ, GM-CSF) during L. longbeachae challenge. Adoptively-transferred MAIT cells maintained an effector-memory cytokine profile in uninfected mice, similar to their corresponding 100 DPI counterparts, with the exception of GM-CSF (compare Figure 4B with Figure 7D). Most strikingly, MAIT cells increased IFN-γ expression significantly by >5 fold from 1.7% in C57BL/6 (Figure 4A, B) to 9.8% (Figure 7C, D), and GM-CSF expression increased 27 fold (3.3% to 9.1%) whilst IL-17 remained largely unchanged (27% to 30%).

Minor comments:

1. Figure 4D and S2C should have statistics added.

Response: We thank the reviewer for noticing this oversight. We have now performed appropriate statistical analysis and updated the figure and figure legend accordingly. Please note that Figure is now numbered as Figure 5D and supplementary Figure S2E.

2. Please switch order of Figure 4E and F, as 4F is described before 4E in manuscript text

Response: We have now made the requested changes. Please note that the Figure is now numbered as Figure 5F, G.

3. Please specify in your title and abstract that you are using Legionella longbeachae, as all the in vivo experiments use L. longbeachae, and not L. pneumophila.

Response: We have now made the requested changes to title and abstract.

4. The authors should cite the Massis et al. paper, as they showed that IFNγ is important for control for L. longbeachae (The Journal of Infectious Diseases, Volume 215, Issue 3, 1 February 2017, Pages 440–451).

Response: We thank the reviewer for bringing this study to our attention. We have now included reference to this work in the discussion section of our revised manuscript.

Specifically, we now state, on page 20, line 425 (of which the latter reference is to the Massis paper)

Furthermore T cell-derived IFN-γ has been shown to stimulate the bactericidal activity of monocyte derived cells in murine L. pneumophila infection¹⁵, and IFN-γ has been shown to be important for the control of L. longbeachae¹⁸.

References

1. Chen, Z. *et al.* Mucosal-associated invariant T-cell activation and accumulation after in vivo infection depends on microbial riboflavin synthesis and co-stimulatory signals. *Mucosal Immunol* **10**, 58-68 (2017).
2. Abdelsamed, H.A., Desai, M., Nance, S.C. & Fitzpatrick, E.A. T-bet controls severity of hypersensitivity pneumonitis. *J Inflamm (Lond)* **8**, 15 (2011).
3. Yoon, H., Legge, K.L., Sung, S.S. & Braciale, T.J. Sequential activation of CD8+ T cells in the draining lymph nodes in response to pulmonary virus infection. *J Immunol* **179**, 391-399 (2007).
4. Kinjyo, I. *et al.* Real-time tracking of cell cycle progression during CD8+ effector and memory T-cell differentiation. *Nature communications* **6**, 6301 (2015).
5. Yoon, H., Kim, T.S. & Braciale, T.J. The cell cycle time of CD8+ T cells responding in vivo is controlled by the type of antigenic stimulus. *PLoS One* **5**, e15423 (2010).
6. Koay, H.F. *et al.* A three-stage intrathymic development pathway for the mucosal-associated invariant T cell lineage. *Nat Immunol* **17**, 1300-1311 (2016).
7. Grimaldi, D. *et al.* Specific MAIT cell behaviour among innate-like T lymphocytes in critically ill patients with severe infections. *Intensive care medicine* (2013).
8. van Wilgenburg, B. *et al.* MAIT cells are activated during human viral infections. *Nature communications* **7**, 11653 (2016).
9. Jiang, J. *et al.* Mucosal-associated Invariant T-Cell Function Is Modulated by Programmed Death-1 Signaling in Patients with Active Tuberculosis. *Am J Respir Crit Care Med* **190**, 329-339 (2014).
10. Loh, L. *et al.* Human mucosal-associated invariant T cells contribute to antiviral influenza immunity via IL-18-dependent activation. *Proc Natl Acad Sci U S A* **113**, 10133-10138 (2016).
11. Becker, T.C., Coley, S.M., Wherry, E.J. & Ahmed, R. Bone marrow is a preferred site for homeostatic proliferation of memory CD8 T cells. *J Immunol* **174**, 1269-1273 (2005).
12. D'Souza, C. *et al.* Mucosal-Associated Invariant T Cells Augment Immunopathology and Gastritis in Chronic Helicobacter pylori Infection. *J Immunol* **200**, 1901-1916 (2018).
13. Ussher, J.E. *et al.* CD161⁺⁺ CD8⁺ T cells, including the MAIT cell subset, are specifically activated by IL-12+IL-18 in a TCR-independent manner. *Eur J Immunol* **44**, 195-203 (2014).
14. Kaech, S.M., Wherry, E.J. & Ahmed, R. Effector and memory T-cell differentiation: implications for vaccine development. *Nat Rev Immunol* **2**, 251-262 (2002).
15. Brown, A.S. *et al.* Cooperation between Monocyte-Derived Cells and Lymphoid Cells in the Acute Response to a Bacterial Lung Pathogen. *PLoS Pathog* **12**, e1005691 (2016).
16. Rahimpour, A. *et al.* Identification of phenotypically and functionally heterogeneous mouse mucosal-associated invariant T cells using MR1 tetramers. *J Exp Med* **212**, 1095-1108 (2015).

17. Meierovics, A., Yankelevich, W.J. & Cowley, S.C. MAIT cells are critical for optimal mucosal immune responses during in vivo pulmonary bacterial infection. *Proc Natl Acad Sci U S A* **110**, E3119-3128 (2013).
18. Massis, L.M. *et al.* Legionella longbeachae Is Immunologically Silent and Highly Virulent In Vivo. *J Infect Dis* **215**, 440-451 (2017).

REVIEWERS' COMMENTS:

Reviewer #1 (Remarks to the Author):

This manuscript by Wang et al presents important new findings regarding the antibacterial function of MAIT cells. Significant components of this paper are the *in vivo* protective effect of MAIT cells in mice against an important human pathogen, the ability of rapid adaptive MAIT cell expansion at a site of infection, the possibility to enhance MAIT cell-mediated protection by vaccination with MR1-presented antigen, and the finding that the *in vivo* protective effect is largely dependent on IFN- γ . Perhaps the most important is the finding that "priming" and expansion of MAIT cells *in vivo* can mediate enhanced protection against infection by a lung pathogen. This finding may have very significant translational implications for human bacterial disease given the broad antibacterial specificity of MAIT cells.

This reviewer has only a couple of minor comments remaining regarding data presentation in figure 2:

First, in Fig 2A it is confusing to use both different colors and different symbols in the same figure, without explaining the meaning of the colors of the bars.

Second, in Fig 2B I doubt that the readers will be able to see anything in the tiny microscopy panels, at least not in a printed copy. Is it possible to focus in on the "CD57/Bl6 3DPI" condition and show the controls in supplement? That way the panels can be made bigger.

A final comment is that the referencing is slightly biased in a patriotic manner, citing mostly Australian and British MAIT cell work. Perhaps look this over.

Reviewer #2 (Remarks to the Author):

The authors have extensively revised the manuscript according to the criticisms raised by the reviewers and I believe the paper is now much stronger, with the BrdU experiment, the added controls for the tetramer histology and for the intracellular stainings for cytokines.

Few minor points:

Line 130: "phagocytosed" bacilli are a push at this magnification.

Line 173: add "in a model of *Francisella tularensis* infection" before ref 14.

Figure 3F: how is the MAIT cell frequency calculated? There is only a two fold increase in MAIT percentage at 5DPI, lower than the 10% shown in panel E. Is it because the dose is 10×10^4 CFU as opposed to 2×10^4 ?

Figure 4: The dots plots now reflect much better the quality of the data. I wonder whether the authors would like to add a comment on the increased MFI in IL17 secretion between naïve mice (suppl. fig) and infected mice (this figure), despite both having similar percentages. I would also add the statistics to the drawings in panels B and C to substantiate the findings. Indeed, although the percentage of IFN γ secretion cells does not greatly increase at 7DPI (is it really significant? Line 228) there is a much greater change in absolute numbers, which becomes relevant for the experiments with IFN γ KO mice.

Figure 5: please change the scale of panel E for consistency with F and G. Line 261, rephrase "in normal C57BL/6.." to WT versus MR1 KO on a C57BL/6 background or something similar.

Line 985: should be Sig S6.

For clarity, I would change the legend in Figures 2A, S1A and B to a layout similar to Fig 1A, as the symbols are too small and do not reflect the colors of the histogram bars.

Reviewer #3 (Remarks to the Author):

The authors have satisfactorily addressed the reviewers' concerns.

REVIEWERS' COMMENTS:

Reviewer #1 (Remarks to the Author):

This manuscript by Wang et al presents important new findings regarding the antibacterial function of MAIT cells. Significant components of this paper are the in vivo protective effect of MAIT cells in mice against an important human pathogen, the ability of rapid adaptive MAIT cell expansion at a site of infection, the possibility to enhance MAIT cell-mediated protection by vaccination with MR1-presented antigen, and the finding that the in vivo protective effect is largely dependent on IFN-gamma. Perhaps the most important is the finding that “priming” and expansion of MAIT cells in vivo can mediate enhanced protection against infection by a lung pathogen. This finding may have very significant translational implications for human bacterial disease given the broad antibacterial specificity of MAIT cells.

This reviewer has only a couple of minor comments remaining regarding data presentation in figure 2:

First, in Fig 2A it is confusing to use both different colors and different symbols in the same figure, without explaining the meaning of the colors of the bars.

We accept the reviewer's comment and have revised the figure to use only a single symbol type.

Second, in Fig 2B I doubt that the readers will be able to see anything in the tiny microscopy panels, at least not in a printed copy. Is it possible to focus in on the “CD57/BL6 3DPI” condition and show the controls in supplement? That way the panels can be made bigger.

We accept the reviewer's comment and have revised Fig 2B to show the merged image at a much larger scale, with only the CD57/BL6 3DPI images in the main manuscript. As requested we have also moved these images and all controls to the online supplement as new Figure S4.

A final comment is that the referencing is slightly biased in a patriotic manner, citing mostly Australian and British MAIT cell work. Perhaps look this over.

We have looked this over: of the 55 references cited 20 are from the USA, 15 from Australia and 8 from France and 4 are from UK which we do not feel represents an unduly strong bias.

Reviewer #2 (Remarks to the Author):

The authors have extensively revised the manuscript according to the criticisms raised by the reviewers and I believe the paper is now much stronger, with the BrdU experiment, the added controls for the tetramer histology and for the intracellular stainings for cytokines.

Few minor points:

Line 130: “phagocytosed” bacilli are a push at this magnification.

In light of the reviewer's comment we have changed this from 'phagocytosed' to 'intracellular'. To assist the reader we now also highlight the relevant cell with a yellow arrow in the figure and have changed the legend to figure 1D by insertion of the following text:

'Yellow arrow: intracellular L. longbechae bacilli.'

Line 173: add “in a model of Francisella tularensis infection” before ref 14.

As requested, we now insert the phrase 'in a model of *Francisella tularensis* infection' before ref 14

Figure 3F: how is the MAIT cell frequency calculated? There is only a two fold increase in MAIT percentage at 5DPI, lower than the 10% shown in panel E. Is it because the dose is 10e4 CFU as opposed to 2 x 10e4?

Murine MAIT cell frequencies are calculated in the same manner throughout: they are the proportion of live, TCRbeta+, CD45+CD19- lymphocytes which are MR1-5-OP-RU tetramer positive. Yes, we did use 10⁴CFU for the BrdU experiments. The mean percentage of MAIT cells at day 3 in this figure is 1.4% and at day 5 is 3.8%, which are indeed both lower than those in Figure 3 panel E. We therefore agree with the reviewer's suggested explanation: the slightly lower MAIT % in Fig 3F was mostly likely due to the lower dose of inoculum used in BrdU assay. The figure legend states the lower inoculum and we believe the method of calculation is now much more apparent to the reader with the addition of the new supplementary figures showing human and murine MAIT cell cytometric gating strategies.

Figure 4: The dots plots now reflect much better the quality of the data. I wonder whether the authors would like to add a comment on the increased MFI in IL17 secretion between naïve mice (suppl. fig) and infected mice (this figure), despite both having similar percentages.

The differences in MFI are apparent to the interested reader from these figures, but given the technical challenges of performing ICS in naïve mice which have very low MAIT cell frequencies, we would not want to over-interpret the data and would rather continue to present only the % or absolute numbers of cytokine-positive cells as shown in the summary graphs of Figures 4B, C.

Nonetheless, in light of the reviewer's comment we now have amended the relevant supplementary figure A and B to show in brackets the proportion of MR1-tetramer positive MAIT cells expressing the cytokine or transcription factor, which will aid the reader in assessing these ratios.

I would also add the statistics to the drawings in panels B and C to substantiate the findings.
We have now added these statistics as requested.

Indeed, although the percentage of IFN γ secretion cells does not greatly increase at 7DPI (is it really significant? Line 228) there is a much greater change in absolute numbers, which becomes relevant for the experiments with IFN γ KO mice.

The interferon secretion is indeed significantly higher at day 7 than uninfected or 100 days post infection, and we now have amended the sentence to state the statistical tests:

IFN- γ secretion was proportionately less, but nevertheless was significantly higher during the acute infection than in cells from uninfected mice or after resolution (ANOVA with post-hoc Tukey's $P=0.03$ and 0.0004 respectively, Figure 4B middle panel)

This may not be immediately apparent from the graph as the axis is optimized for presentation of the non-MAIT T cell subsets as well, but the large 'n' with very little variance in the data at each time point make these differences significant.

Figure 5: please change the scale of panel E for consistency with F and G.

We have revised this figure as requested.

Line 261, rephrase "in normal C57BL/6.." to in WT versus MR1 KO on a C57BL/6 background or something similar.

As requested we have rephrased as follows:

In wild type versus MRI^{-/-} on a C57BL/6 background we observed...

Line 985: should be Sig S6.

We are grateful for the correction, which has been made.

For clarity, I would change the legend in Figures 2A, S1A and B to a layout similar to Fig 1A, as the symbols are too small and do not reflect the colors of the histogram bars.

As requested we have made the suggested change to Figure 2A, S1A and S1B which now have a layout similar to Fig 1A.

Reviewer #3 (Remarks to the Author):

The authors have satisfactorily addressed the reviewers' concerns.